# 🌿 VERA: A General-Purpose Plausibility Estimation Model for Commonsense Statements

**Jiacheng Liu**[♡*]  **Wenya Wang**[♡♣*]  **Dianzhuo Wang**[◇]
**Noah A. Smith**[♡♠]  **Yejin Choi**[♡♠]  **Hannaneh Hajishirzi**[♡♠]

[♡]Paul G. Allen School of Computer Science & Engineering, University of Washington
[♠]Allen Institute for Artificial Intelligence  [♣]Nanyang Technological University
[◇]John A. Paulson School of Engineering and Applied Sciences, Harvard University
liujc@cs.washington.edu  [*]Equal contribution.

## Abstract

Today's language models can be remarkably intelligent yet still produce text that contains trivial commonsense errors. Therefore, we seek a retrospective verification approach that can reflect on the commonsense plausibility of the machine text, and introduce VERA, a general-purpose model that learns to estimate the commonsense plausibility of declarative statements. To support diverse commonsense domains, VERA is trained on ∼7M commonsense statements that are automatically converted from 19 QA datasets and two commonsense knowledge bases, and using a combination of three training objectives. When applied to solving commonsense problems in the verification format, VERA substantially outperforms existing models that can be repurposed for commonsense verification, even including GPT-3.5/ChatGPT/GPT-4, and it further exhibits generalization capabilities to unseen tasks and provides well-calibrated outputs. We find that VERA excels at filtering machine-generated commonsense knowledge and is useful in detecting erroneous commonsense statements generated by models like ChatGPT in real-world settings.

## 1 Introduction

We introduce VERA, a general-purpose commonsense statement verification model. This model is designed to estimate the plausibility of declarative, natural language statements based on commonsense knowledge.

We build VERA in response to the absence of good detectors of commonsense errors in text generated by language models (LMs). LMs have been advancing rapidly and have demonstrated remarkable success in various tasks, including question answering, natural language inference, sequence classification, and text generation. Yet these models still make simple commonsense mistakes. As shown in Figure 1, as of February 23, 2023, ChatGPT (OpenAI, 2022a) reportedly output the text

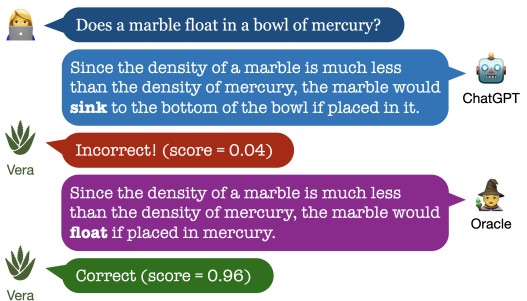

Figure 1: VERA estimates the correctness of declarative statements. Example adapted from a contribution made by Henry Minsky to Marcus and Davis (2023) on February 23, 2023.

*"since the density of a marble is much less than the density of mercury, the marble would sink to the bottom of the bowl if placed in it"*, which is clearly incorrect. This kind of failure raises concerns about the reliability and trustworthiness of these models (Lin et al., 2022).

VERA estimates a plausibility score for a commonsense statement based on its commonsense knowledge about the world. It contrasts with *fact* verification methods (Thorne et al., 2018; Wadden et al., 2020), which verify the correctness of claims based on evidence from a text corpus. VERA enables plausibility estimation where direct evidence is often not retrievable from some corpus, and usually some implicit, fuzzy reasoning is needed. It operates solely with the commonsense knowledge stored in its model parameters, and does not have a retrieval component.

VERA is built on top of T5 (Raffel et al., 2020), a generic pretrained LM, by finetuning on a vast collection of correct and incorrect commonsense statements sourced from knowledge bases (KBs) and question answering (QA) datasets. The 21 data sources (Table 5, appendix) amount to ∼7M statements encompassing a wide spectrum of domains, including general, scientific, physical, and social commonsense, as well as quantitative (reasoning about numbers) and qualitative (reasoning about

qualitative relationships such as *smaller*) commonsense. We propose a novel two-stage training process that takes into account the scale and quality of data from different sources. In addition to the standard multiple-choice binary classification objectives, we adopt a supervised contrastive loss (Khosla et al., 2020) to magnify the distinction between similar statements with different correctness labels. Furthermore, we propose an automatic way of augmenting the training data by eliciting LMs to generate incorrect answers to commonsense questions and empirically find it helps generalization.

We evaluate VERA in the following applications:

- **Excelling in commonsense problems over GPT-series when repurposed for verification (§5.1).** VERA can be applied to solve multiple-choice and boolean commonsense problems when expressed in the verification format, by scoring and ranking candidate hypotheses. It substantially outperforms existing models repurposed for commonsense verification (including GPT-3.5, ChatGPT and GPT-4), improving upon the best existing baseline, Flan-T5, with absolute improvement of 6% on seen benchmarks and 4% on unseen ones.
- **Filtering LM-generated commonsense knowledge (§5.2).** VERA can filter noisy commonsense knowledge statements generated by other LMs, improving the effectiveness of LM-generated knowledge in downstream knowledge-augmented inferences. VERA is well-calibrated, enabling filtering at customized thresholds.
- **Detecting commonsense errors in ChatGPT outputs (§5.3).** Through a preliminary analysis, we find that VERA can identify commonsense errors made by ChatGPT in-the-wild, with a precision of 91% and a recall of 74%. An example of VERA in action is shown in Figure 1.

We hope VERA can be a useful tool for improving the commonsense correctness of existing generative LM output and inspire more effort toward general-purpose and robust verification methods.

## 2   Problem Definition and Scope

Our goal is to build a model that can estimate the plausibility of any given *commonsense statement*. The model takes as input a statement that (1) is expressed in **natural language**; (2) is **declarative**,

as opposed to interrogative questions; (3) is **self-contained** without requiring additional context to comprehend; (4) has an objective, binary **correctness label**; and (5) in principle can be labeled using widely-held **commonsense knowledge** about the world. Encyclopedic knowledge (e.g., *Ljubljana is the capital of Slovenia.*) is out of scope. Moving forward, unless explicitly noted, we use *commonsense statement* to refer to statements within the above scope. Though somewhat strict, this scope covers a broad range of potential applications.

For an input commonsense statement $x$, the model should output a real-valued score $s \in [0, 1]$ that represents its estimated plausibility of $x$. While the gold correctness label is binary, we let the model output a score to reflect its confidence. A score of 1.0 means that it is completely confident that $x$ is correct, and a score of 0.0 means it is completely confident that $x$ is incorrect. When predicting correctness label from the score, we use 0.5 as the threshold.

## 3   Method

In this section, we describe the whole pipeline to build VERA. We start from curating large-scale training data including both correct and incorrect statements from diverse commonsense tasks (§3.1). Next, we learn a scoring model that takes a statement and returns a continuous score by finetuning a LM via 3 training objectives (§3.2). An additional post hoc calibration strategy is applied to make the output scores well-calibrated (§3.3).

### 3.1   Data Construction

Labeled commonsense statements usually do not appear in text in the wild, while some commonsense question answering (QA) datasets and commonsense knowledge bases (KBs) are good sources for this kind of statements. We collect correct and incorrect commonsense statements from the above two types of data source. Table 1 shows some examples on how these statements can be converted from QA problems and KB entries. In total, we obtain ∼7M statements (for training) from 19 QA datasets (§3.1.1) and two KBs (§3.1.2) that encompass a wide spectrum of commonsense domains. Table 5 (appendix) lists these datasets with statistics. All datasets we use are publicly available.

### 3.1.1   From Commonsense QA Datasets

Numerous commonsense reasoning datasets have been published in recent years (Davis, 2023), and

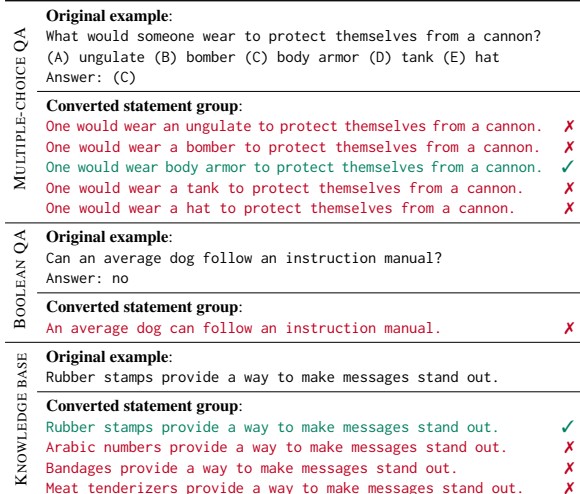

| | | |
|---|---|---|
| **MULTIPLE-CHOICE QA** | **Original example:**
`What would someone wear to protect themselves from a cannon?`
`(A) ungulate (B) bomber (C) body armor (D) tank (E) hat`
`Answer: (C)` | |
| | **Converted statement group:**
`One would wear an ungulate to protect themselves from a cannon.` ✗
`One would wear a bomber to protect themselves from a cannon.` ✗
`One would wear body armor to protect themselves from a cannon.` ✓
`One would wear a tank to protect themselves from a cannon.` ✗
`One would wear a hat to protect themselves from a cannon.` ✗ | |
| **BOOLEAN QA** | **Original example:**
`Can an average dog follow an instruction manual?`
`Answer: no` | |
| | **Converted statement group:**
`An average dog can follow an instruction manual.` ✗ | |
| **KNOWLEDGE BASE** | **Original example:**
`Rubber stamps provide a way to make messages stand out.` | |
| | **Converted statement group:**
`Rubber stamps provide a way to make messages stand out.` ✓
`Arabic numbers provide a way to make messages stand out.` ✗
`Bandages provide a way to make messages stand out.` ✗
`Meat tenderizers provide a way to make messages stand out.` ✗ | |

Table 1: Conversions from original commonsense QA problems and knowledge base entries to statement groups that are used for training.

many of them are in the format of multiple-choice QA (selecting the correct answer out of a set of choices) or boolean (yes/no) QA. These can be easily converted to correct and incorrect commonsense statements. From multiple-choice QA problems, we combine the question and each answer choice to form declarative statements, which are correct when using the correct answer, and incorrect otherwise. From boolean QA problems, we convert the question into a declarative statement, and keep the original label as the correctness label. Concrete examples can be found in Table 1.

**Statement groups.** We refer to statements originating from the same problem as a *statement group*. Note that statement groups originating from multiple-choice problems contain at least two statements, of which one and only one is correct; statement groups originating from boolean problems contain only one statement, and it can be either correct or incorrect.

We do conversion to declarative statements automatically. From QA datasets, we create declarative statements from QA problems using the following method:

- If the problem contains a question, we convert the question and choice into a declarative statement using the question conversion model created by Chen et al. (2021).

- If the question is cloze-style, we replace the blank with the choice.

- If the question is an incomplete sentence and

the choice is a continuation to it, we concatenate the question and the choice.

- If there is no question and the problem only asks to choose between some choices, we use the choice as the declarative statement.

- For boolean problems, we always use *yes* as the choice and create a single declarative statement for each problem. We use the original label as the correctness label of this statement.

In total, 19 commonsense QA datasets contribute ∼200k statement groups and ∼400k statements to the training set of VERA.

**LM-augmented falsehoods.** Existing commonsense QA datasets are mostly manually constructed or assembled from standard school exams. A model trained on these datasets might overfit specific annotation patterns from humans which may limit generalization. Therefore, we augment QA problems with LM-generated answers and construct additional incorrect statements. Specifically, for a multiple-choice question, we use a small LM to sample 50 possible answers to the question, and select the 3 least probable answers with generation probability less than 0.15 (making these unlikely to be correct answers). This threshold is chosen based on manual inspection over a small portion of examples. We observe generated answers with probability larger than 0.15 are more likely to be plausible. We create LM-augmented falsehoods for the training set of 9 commonsense QA datasets, as noted in Table 5 (appendix).

### 3.1.2 From Commonsense KBs

Commonsense KBs (e.g., Atomic2020 in Hwang et al. (2020), and GenericsKB in Bhakthavatsalam et al. (2020)) contain a large number of correct commonsense statements. To create incorrect statements, we automatically perturb KB entries by replacing the subject with three random subjects that appear in the KB. Table 1 shows how to convert an entry in GenericsKB to a statement group containing four statements, three of which are augmented via perturbations. The perturbed statements are relatively easy to identify and may contain false negatives. As noted in §3.2.4, we use these KB-constructed statements in a separate training stage that precedes training with QA-constructed statements. In total, two commonsense KBs contribute ∼1.6M statement groups and ∼6M statements to the training set of VERA.

### 3.2 Model Training

#### 3.2.1 Model Architecture

Given a statement $x$, VERA outputs a real-valued score $s \in [0, 1]$. As we will use a transformer-based LM as the backbone of VERA, we first extract the input representation $\mathbf{h}$ by selecting the last hidden state corresponding to the EOS input token. We choose EOS because it is capable to encode the entire input in both bidirectional encoder models (e.g., T5's encoder) and left-to-right decoder models (e.g., LLaMA). Then a linear layer projects $\mathbf{h}$ to a scalar logit $z$, followed by a sigmoid function $\sigma(\cdot)$ that transforms the logit into a score $s$. Formally,

$$\mathbf{h} = f_{\text{LM}}(x), z = f_{\text{linear}}(\mathbf{h}), s = \sigma(z).$$

For brevity, we use $\mathbf{h}(x)$, $z(x)$ and $s(x)$ to refer to the representation, logit and score of an arbitrary input $x$.

#### 3.2.2 Batching

The data we construct consists of statements belonging to different statement groups. For reasons we will describe in §3.2.3, we put all statements belonging to the same statement group into the same batch. Each batch may contain multiple complete statement groups. We denote by $B_G$ the number of statement groups and $B_S$ the number of statements in total within a single batch. We denote the statement groups as $\{X_j\}_{j=1}^{B_G}$, and the statements as $\{x_i\}_{i=1}^{B_S}$. $\{X_j\}_{j=1}^{B_G}$ is a partition of $\{x_i\}_{i=1}^{B_S}$. $y_i \in \{0, 1\}$ is the correctness label of $x_i$.

#### 3.2.3 Training Objectives

The model is trained with a linear combination of three losses: a binary classification loss, a multi-class loss, and a supervised contrastive loss, $\mathcal{L} = \alpha \mathcal{L}_{\text{bin}} + \beta \mathcal{L}_{\text{mc}} + \gamma \mathcal{L}_{\text{ctr}}$, which we describe below.

**Binary classification loss.** Naively, commonsense statement verification can be viewed as a binary classification task. Under this setting, the loss is

$$\mathcal{L}_{\text{bin}} = -y_i \log s(x_i) - (1 - y_i) \log(1 - s(x_i)).$$

**Multi-class loss.** We expect the model to be robust against nuances in commonsense statements. Ideally, the model should be able to recognize opposite correctness labels for a group of seemingly similar statements in surface forms, such as statements created from different choices of the same question, or perturbed from the same piece of knowledge in

a KB. To achieve this goal, we treat each statement group as a multi-class classification problem, maximizing the log-likelihood of the single correct statement in the statement group after passing the logits through a softmax. Formally,

$$\mathcal{L}_{\text{mc}} = -\log \frac{\exp z(x_{j*})}{\sum_{c=1}^{C_j} \exp z(x_{jc})},$$

where $x_{j*}$ is the correct statement in $X_j$. Note that the multi-class loss is not applicable to statement groups with only one statement (i.e., statement groups from boolean QA datasets). We empirically find that the multi-class loss indeed improves generalization towards unseen multiple-choice QA datasets as indicated in Figure 3 (appendix).

**Supervised contrastive loss.** It has been shown (Khosla et al., 2020) that supervised contrastive learning helps to improve model robustness and generalization against input variations. In light of this, we further adopt supervised contrastive learning on top of the input representations $\mathbf{h}$. We show in Figure 3 (appendix) that the contrastive loss indeed improve generalization to unseen datasets. For each anchor statement $x_i$ in a batch, the contrastive loss aims to maximize the similarity between $x_i$ and each other statement $x_p$ that has the same correctness label as $x_i$ (i.e., positive example). At the same time, we push apart $x_i$ and other statements $x_n$ that has opposite correctness label as $x_i$ (i.e., negative example). The supervised contrastive loss is

$$\mathcal{L}_{\text{ctr}} =$$
$$-\log \frac{\sum_{k \in \mathcal{P}(i)} \exp[\cos(\mathbf{h}(x_i), \mathbf{h}(x_k))/\tau]}{\sum_{k \in \mathcal{P}(i) \cup \mathcal{N}(i)} \exp[\cos(\mathbf{h}(x_i), \mathbf{h}(x_k))/\tau]},$$

where $\tau$ is a temperature hyperparameter, $\cos(\cdot, \cdot)$ refers to cosine similarity, $\mathcal{P}(i) \subseteq [B_S]$ is the index set of statements that are positive examples for $x_i$, and $\mathcal{N}(i) \subseteq [B_S]$ is the index set of statements that are negative examples for $x_i$. Formally,

$$\mathcal{P}(i) = \{k \mid 1 \le k \le B_S, y_k = y_i, k \ne i\},$$
$$\mathcal{N}(i) = \{k \mid 1 \le k \le B_S, y_k \ne y_i\}.$$

#### 3.2.4 Two-Stage Training

Since data sourced from KBs are larger in scale but more noisy than data sourced from QA datasets, we take a two-stage training approach. In training stage A, we start from a pre-trained LM and train with data sourced from KBs. In training stage B,

we start from the model obtained in stage A and train with data sourced from QA datasets. During experiments we found that this setting is better than single-stage training with either data source or a mixture of the two.

### 3.3 Inference and Calibration

An ideal plausibility estimation model should be calibrated, that is, its confidence in its predictions should be approximately equal to the actual frequency of correctness. During early experiments, we found that VERA tends to be overconfident. Therefore, we apply a post hoc calibration on VERA's output. Following the temperature scaling method introduced in Guo et al. (2017), during inference we divide the model-predicted logit by a temperature $T$ before computing the score, that is,

$$\mathbf{h} = f_{\text{LM}}(x), z = f_{\text{linear}}(\mathbf{h}), \tilde{z} = z/T, s = \sigma(\tilde{z}).$$

Note that no temperature scaling is applied during model training.

With predictions on a validation set $\mathcal{D} = \{(x_i, y_i)\}_{i=1}^{\mathcal{D}}$, we estimate $T$ that gives the minimal expected calibration error (ECE) (Naeini et al., 2015) on this validation set. Equation 1 in §C.1 shows how ECE is computed. In practice, we use the combined development sets of the seen datasets (§4.2) to estimate $T$, and the optimal $T$ becomes a parameter of VERA. Note that temperature scaling does not change the relative ordering of prediction scores, and thus the other performance metrics (e.g., accuracy) are not affected (see detailed explanation in §B.2).

## 4 Experimental Setup

In this section, we provide more details of model training, the evaluation protocol and metrics, and describe the baseline models we benchmark.

### 4.1 Training Details

**Datasets.** For training stage A, we use the ∼1.6M statement groups (∼6M statements) sourced from two commonsense KBs; for training stage B, we use the ∼200k statement groups (∼400k statements) sourced from 19 commonsense QA datasets. For each training stage, we mix the training sets of all datasets together, without any re-weighting.

**Models.** We use two types of pretrained LMs as the backbone of VERA: (1) the encoder of T5 (Raffel et al., 2020), which is a bidirectional encoder model; (2) LLaMA (Touvron et al.,

2023), which is a left-to-right decoder model. For the T5 encoder, we start from the pretrained `T5-v1.1-XXL`[1] whose encoder has about 5B parameters, and refer to the resulting model as VERA-T5. (During experiments we found that starting from `Flan-T5-XXL`[2] performs slightly worse than starting from `T5-v1.1-XXL`.) For LLaMA, we start from the pretrained `LLaMA-7B` and refer to the resulting model as VERA-LLaMA. As we will see, VERA-T5 has better performance than VERA-LLaMA, so unless explicitly specified, when we say VERA we mean VERA-T5. See Table 8 (appendix) for the complete hyperparameter settings and §C for the implementation details.

### 4.2 Evaluation and Baselines

**Evaluation protocol.** We divide our evaluation into two parts: (1) *Seen* benchmarks, whose training set is used for model training. (2) *Unseen* benchmarks, whose training set is not used for model training. We futher divide up the unseen benchmarks into *type 1* and *type 2*, where in type 1 benchmarks the task is similar to those in the seen benchmarks, while type 2 benchmarks are a bit further away in terms of the nature of the task. Examples of type 2 unseen benchmarks include HellaSwag which is contextualized with event descriptions, and CREAK which involves reasoning among different entities.

Depending on the nature of the evaluation benchmark, we use different metrics to evaluate our model's performance. Unless explicitly said otherwise, we report performance on the development set, where the gold labels are available, and we do not use the development sets of unseen datasets for model selection. The overall metric reported over multiple benchmarks is the unweighted average of the metric over all these benchmarks, which accounts for the differently-sized evaluation sets.

**Metrics.** We report accuracy for multiple-choice and balanced boolean benchmarks. For those unbalanced boolean benchmarks (e.g., LM-generated knowledge filtering datasets), we report area under the ROC curve (AUROC) and average precision (AP). To measure how well the model-predicted scores reflect confidence, we measure the ECE (Naeini et al., 2015) on the boolean benchmarks, following Equation 1.

---

[1] https://huggingface.co/google/t5-v1_1-xxl
[2] https://huggingface.co/google/flan-t5-xxl

| Accuracy | Seen | Unseen (type 1) | Unseen (type 2) |
|---|---|---|---|
| SKD Critic (355M) | 36.64 | 38.34 | 43.40 |
| I2D2 Critic (355M) | 55.03 | 54.79 | 67.11 |
| UnifiedQA-v2 (11B) | 56.33 | 59.73 | 53.95 |
| Entailer (11B) | 73.79 | 71.47 | 70.72 |
| GPT-3.5 (175B) | 75.41 | 71.03 | 78.87 |
| ChatGPT[†] | 62.11 | 61.20 | 62.83 |
| GPT-4[†] | 72.35 | 77.40 | 70.29 |
| Flan-T5[‡] (11B) | 79.50 | 77.62 | 78.89 |
| VERA-LLaMA (7B) | 82.99 | 75.51 | 82.56 |
| VERA-T5 (5B) | **85.51** | **81.65** | **83.37** |

Table 2: Results on problem-solving with VERA on seen and unseen benchmarks. Average accuracy on the development sets is reported. Accuracy across different parts (seen, unseen (type 1), unseen (type 2)) are not directly comparable due to different underlying benchmarks. See Figure 5 and Table 9, 10, 11 (appendix) for full results. †: The performance of ChatGPT and GPT-4 may be under-estimated because we don't have access to the raw token logits. ‡: Flan-T5 has been trained on some unseen benchmarks we use; see Table 7 (appendix) for details on data contamination.

**Baseline Models.** We compare VERA with the best publicly available models that can be directly used or repurposed for commonsense statement verification. Roughly in increasing order of performance, these models are: SKD Critic (West et al., 2021), I2D2 Critic (Bhagavatula et al., 2022), UnifiedQA-v2 (Khashabi et al., 2022), Entailer (Tafjord et al., 2022), GPT-3.5 (OpenAI, 2022b), ChatGPT (OpenAI, 2022a), GPT-4 (OpenAI, 2023), and Flan-T5 (Chung et al., 2022). See more details in §C.2.

## 5 Evaluation Results

In this section, we evaluate the ability of VERA to estimate the plausibility of commonsense statements and compare it with the baseline models. We show the effectiveness of VERA in three scenarios: solving commonsense problems, filtering LM-generated commonsense knowledge, and detecting commonsense errors in ChatGPT outputs.

### 5.1 Solving Multiple-Choice and Boolean Commonsense Problems

The output plausibility scores from VERA can be used for solving multiple-choice and boolean commonsense problems. We first convert the problems into the statement group format (§3.1). For multiple-choice problems, we choose the statement with the highest score in the statement group. For boolean problems, we use $s = 0.5$ as the threshold

to predict correctness labels of statements.

Table 2 reports the results when VERA is applied to solve commonsense problems. See Figure 5 and Table 9, 10, 11 (appendix) for full results including AUROC and AP. On seen benchmarks (16 multiple-choice and one boolean), VERA outperforms the best baseline, Flan-T5, by 6% on (absolute) accuracy and 9% on AUROC. VERA beats Flan-T5 by 4% accuracy and 5% AUROC on type 1 unseen benchmarks (four multiple-choice and one boolean), and by 4% accuracy and 6% AUROC on type 2 unseen benchmarks (five multiple-choice and two boolean), demonstrating good generalization. VERA-T5 has better performance than VERA-LLaMA across the board, which may be due to its bidirectional connectivity. Aside from performance, VERA also has good calibration, with ECE no higher than 3% on seen and unseen benchmarks. The post hoc calibration method improves calibration across all three parts.

Typically we may need to choose a threshold for binary classification in boolean datasets. However, we notice that a zero logit ($z = 0$) is generally close to the optimal decision threshold between correct and incorrect commonsense statements. Therefore we do not estimate a model-specific threshold, and simply use the default threshold: $z = 0$, or equivalently, $s = 0.5$.

### 5.2 Filtering LM-generated Commonsense Knowledge

Figure 2 reports the results when VERA is applied to filter LM-generated commonsense knowledge. On the two seen benchmarks, SKD_anno and I2D2_anno, VERA is a better knowledge filter than all baseline models, in terms of both AUROC and AP. In particular, on I2D2_anno it outperforms the I2D2 critic model by 2% AUROC, which is specifically trained on the I2D2_anno dataset and does not generalize well to other benchmarks. On the unseen benchmark, Rainier_anno, VERA is also comparable with the best baselines like Flan-T5 and GPT-3.5. As for calibration, the ECE is no higher than 8% on all three benchmarks.

We find that filtering commonsense knowledge using VERA can greatly improve the performance of knowledge-augmented reasoning methods. In the Generated Knowledge Prompting framework (Liu et al., 2021), when solving a commonsense QA problem, first a knowledge model generates several commonsense knowledge statements rele-

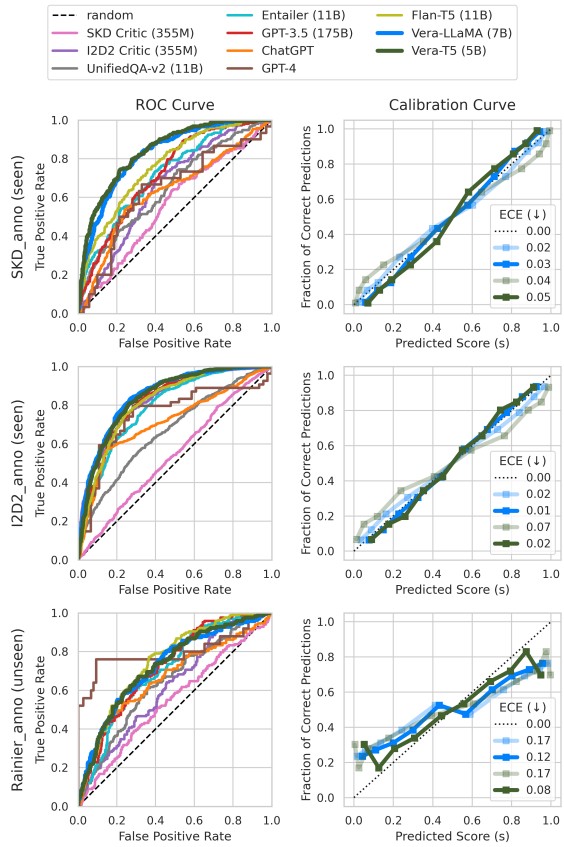

Figure 2: Results for filtering LM-generated commonsense knowledge with VERA. We plot the calibration curve for both the uncalibrated version (w/ faded color) and calibrated version (w/ saturated color) of the VERA model. Results on the development sets are reported. See Figure 6 for full results.

| Generator | Filter | QA Model | Acc | Usefulness | Δ |
|---|---|---|---|---|---|
| – | – | UnifiedQA | 60.45 | – | – |
| GPT-3 | – | UnifiedQA | 67.44 | +6.99 | – |
| GPT-3 | VERA | UnifiedQA | **70.67** | **+10.22** | **+46%** |
| – | – | UnifiedQA | 60.45 | – | – |
| Rainier | – | UnifiedQA | 61.78 | +1.33 | – |
| Rainier | VERA | UnifiedQA | **64.88** | **+4.43** | **+233%** |

Table 3: Results of introducing VERA into the Generated Knowledge Prompting pipeline (Liu et al., 2021). The QA model is `UnifiedQA-large`, and the generator is either `GPT-3 (davinci)` or `Rainier-large` when applicable. Average accuracy on the development set is reported; see Table 12 (appendix) for detailed results.

usefulness of GPT-3's and Rainier's knowledge by 46% and 233%, respectively. VERA can effectively supervise and improve the quality of commonsense knowledge generated by a much larger model, `GPT-3 (davinci)`. Detailed results (Table 12, appendix) show that there is increased effectiveness in every individual benchmark.

## 5.3 Preliminary Study on Detecting Commonsense Errors made by ChatGPT

VERA can be useful in detecting commonsense mistakes made by generative LMs in-the-wild. We collected 27 anecdotes from the Internet where people reported ChatGPT making commonsense errors, and manually rewrote them into their correct versions, obtaining 54 statements in total.

When detecting incorrect commonsense statements in this dataset, VERA has a precision of 91% and a recall of 74%, amounting to an $F_1$ score of 82%. Table 4 shows how VERA scores some of these these erroneous commonsense statements and their manually corrected version. In 7 out of the 9 cases, VERA assigns a low score to the original, incorrect statement, and a high score to the corrected statement. For example, *"since the density of a marble is much less than the density of mercury, the marble would sink to the bottom of the bowl if placed in it"* receives a score of 0.04 and is identified as an incorrect statement, whereas *"since the density of a marble is much less than the density of mercury, the marble would float if placed in mercury"* receives a score of 0.96 and is identified as a correct statement. Meanwhile, there are also some failure cases. VERA believes that *"it is possible for a solar eclipse to be followed by a lunar eclipse the next day"*, and fails to reject that *"it is possible to draw a diagonal line in a triangle"*.

vant to the question, and then a QA model makes predictions based on them. A big problem that hinders the effectiveness of this framework is that model-generated knowledge is not always factual, and incorrect knowledge statements can mislead the QA model. We introduce VERA to filter these statements before passing them to the QA model. In particular, we keep those statements that receive a score higher than 0.5 from VERA.

Following Liu et al. (2022b), we use `UnifiedQA-large` as the QA model, and consider two knowledge models: few-shot `GPT-3 (davinci)` (Brown et al., 2020) and `Rainier-large` (Liu et al., 2022b). We follow the evaluation settings as in Liu et al. (2022b), and for few-shot `GPT-3 (davinci)`, we use the same task-specific few-shot prompts and same process to generate silver knowledge as in Liu et al. (2022b). Results are shown in Table 3. Applying knowledge filtering with VERA increases the

| Date | Original / Corrected | Score | Pred |
|---|---|---|---|
| 2023/01/05 | It is possible for a solar eclipse to be followed by a lunar eclipse the next day. | 0.86 | ✓ |
| | It is impossible for a solar eclipse to be followed by a lunar eclipse the next day. | 0.48 | ✗ |
| 2023/01/06 | The time it takes for a given number of cars to travel a fixed distance is directly proportional to the number of cars. | 0.26 | ✗ |
| | The time it takes for a given number of cars to travel a fixed distance is invariant of the number of cars. | 0.52 | ✓ |
| 2023/01/06 | If A sits next to B and B sits next to C, then A must sit next to C. | 0.20 | ✗ |
| | If A sits next to B and B sits next to C, then A may not sit next to C. | 0.60 | ✓ |
| 2023/01/10 | If two cats can eat two cans of food in a minute, then it would take six cats to eat three cans of food in a minute. | 0.05 | ✗ |
| | If two cats can eat two cans of food in a minute, then it would take three cats to eat three cans of food in a minute. | 0.67 | ✓ |
| 2023/01/11 | A three-dimensional cube has eight faces. | 0.46 | ✗ |
| | A three-dimensional cube has six faces. | 0.70 | ✓ |
| 2023/01/30 | It is possible to draw a diagonal line in a triangle. | 0.80 | ✓ |
| | It is impossible to draw a diagonal line in a triangle. | 0.28 | ✗ |
| 2023/02/21 | 70 is a smaller number than 58. | 0.14 | ✗ |
| | 70 is a larger number than 58. | 0.85 | ✓ |
| 2023/02/23 | Since the density of a marble is much less than the density of mercury, the marble would sink to the bottom of the bowl if placed in it. | 0.04 | ✗ |
| | Since the density of a marble is much less than the density of mercury, the marble would float if placed in mercury. | 0.96 | ✓ |
| 2023/02/25 | Both a house and a pound of feathers weigh the same, which is one pound. | 0.25 | ✗ |
| | A house weighs more than one pound, while a pound of feathers weighs one pound. | 0.87 | ✓ |

Table 4: Examples of commonsense mistakes made by ChatGPT, and how VERA can detect them. In each section, the first line is the original, incorrect commonsense statement in ChatGPT's output, and the second line is the authors' manually corrected version of the statement. Each statement is followed by VERA's score and predicted correctness label. Examples are adapted from Venuto (2023); Marcus and Davis (2023); Borji (2023).

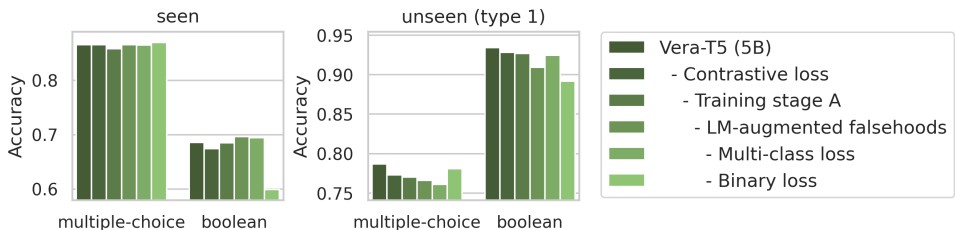

Figure 3: Ablation results. Average accuracy on the development sets is reported. Components are incrementally removed from the training process, except for the multi-class loss and the binary loss; the hierarchy is indicated in the legend.

## 5.4 Analysis

**Ablations.** We conduct an ablation study by incrementally removing the following components from the training process: contrastive loss (§3.2.3), training stage A (§3.2.4), LM-augmented falsehoods (§3.1), multi-class loss or binary loss (§3.2.3). Since at least one of the multi-class loss and the binary loss is needed, we remove them separately and observe the effect of training with a single loss.

Results are shown in Figure 3. Overall, the ablated components have more impact on unseen benchmarks than seen ones. Removing the contrastive loss hurts performance mostly on unseen datasets, implying that the contrastive objective is beneficial for generalization. Removing training stage A hurts performance across the board, emphasizing the importance of training with large-scale commonsense knowledge. LM-augmented falsehoods are most helpful on unseen benchmarks, with a little sacrifice in the performance on seen benchmarks. The multi-class loss is most helpful on multiple-choice benchmarks, while removing the binary loss substantially hurts performance on boolean benchmarks.

**Scaling Trends of VERA.** We trained variants of VERA that are based on smaller versions of the T5 encoder, and show the results in Figure 4. Model performance increases steadily with size, and does not show evidence of saturation at 5B parameters, suggesting that better commonsense plausibility estimation models might be yielded from larger pretrained LMs.

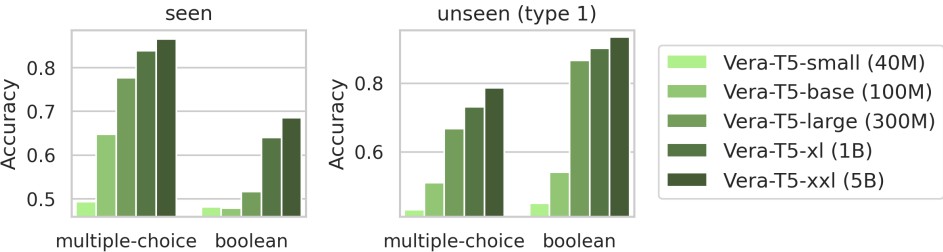

Figure 4: Scaling trends of commonsense statement verifiers.

**Format: Verification vs. QA.** In this paper, we focus on the verification format to solve commonsense problems. A comprehensive discussion on how this format compares with the QA format is provided in §E and Figure 7.

## 6 Related Work

**Commonsense verifiers.** Prior work has explored the idea of verifying commonsense statements. SYMBOLIC KNOWLEDGE DISTILLATION (West et al., 2021) and I2D2 (Bhagavatula et al., 2022) train models to classify the acceptability of model-generated commonsense statements. The ENTAILER (Tafjord et al., 2022) model is partially trained to score the validity of a given hypothesis. These models are trained on relatively small-scale, domain-specific data and do not generalize well to broader commonsense domains. Some other work uses pretrained LMs with few-shot prompting to verify commonsense statements (Kadavath et al., 2022; Jung et al., 2022). In this work, we develop a general-purpose commonsense statement verifier that works out-of-the-box in zero-shot setting.

**Verification in other tasks.** Beyond commonsense statements, the problem of verification has been extensively studied on various NLP tasks. NLI (Liu et al., 2019, 2022a; Zhang et al., 2017) can be viewed as an *entailment verification* task. Chen et al. (2021) presents a method for *QA verification* by transforming the context passage and question-answer pair into a premise-hypothesis format as in NLI. Some work build models to perform *reasoning verification* – classifying whether a premise supports or refutes a hypothesis (Bostrom et al., 2022; Sprague et al., 2022; Yang et al., 2022; Tafjord et al., 2022). On the other hand, *fact verification* (Thorne et al., 2018; Wadden et al., 2020) requires judging the validity of claims against a corpus of evidence (e.g., Wikipedia). These tasks feature context-sensitive or knowledge-intensive hy-

potheses to verify and are typically complemented with additional context. In contrast, we focus on verifying standalone commonsense statements where no context is required or provided.

**Generation vs. verification.** With the rapid progress in generative LMs, researchers have been largely building general-purpose problem-solving methods with a generative approach (Khashabi et al., 2020, 2022; Lourie et al., 2021; Tafjord and Clark, 2021; Wei et al., 2022). However, current generative LMs are still prone to hallucination errors and lack an intrinsic mechanism to express confidence level on their outputs. Verification, on the other hand, shows promise to complement these shortcomings and has been adopted to improve the outcome of generation (Chen et al., 2021; Jiang et al., 2022). In this work, we take a pure verification approach and build a general-purpose verifier for commonsense statements, which to our best knowledge is the first of its kind.

## 7 Conclusion and Future Work

We introduced VERA, a general-purpose verification model for commonsense statements and an early step toward tools for mitigating commonsense errors in text generated by language models. VERA achieves state-of-the-art performance when solving commonsense problems in the verification format, excels at filtering LM-generated commonsense knowledge statements, and is found useful in detecting erroneous commonsense statements from generative LMs. Furthermore, the scores produced by VERA are well-calibrated; and could be used for plausibility score estimation for declarative statements if needed. As VERA mainly targets on single-sentence statements, future work may consider verification of multi-sentence or long-form statements, or contextualized/defeasible commonsense statements.

## Limitations

VERA aims, and is trained, to predict the plausibility of statements based on objective commonsense knowledge of our world. It is not intended to handle text outside the scope of commonsense statements (e.g., encyclopedic facts, reading comprehension with fictional worlds). It is not trained or evaluated on moral commonsense data, so its capability of making moral predictions is unknown. It gives a prediction even if the input falls out of its intended scope, which could be mitigated by an additional scope guard to determine its applicability. In addition, it is not trained to handle very long and compositional input. Although greatly outperforming existing systems, VERA is not perfect and may make incorrect predictions. It is not very robust under syntactic variations of the input, such as paraphrases and negations. As the training data may contain bias or toxicity, VERA may also make predictions that are perceived as ethically problematic. The output of VERA does not reflect the authors' view. VERA is a research prototype, and it is not designed for making real-world decisions.

## Acknowledgments

We thank Sean Welleck, Peter West, Alisa Liu, Jaehun Jung, Chandra Bhagavatula, Ram Pasunuru, Asli Celikyilmaz, and members of the H2lab, Xlab and ARK lab for their discussion and constructive feedback. This work was funded in part by the DARPA MCS program through NIWC Pacific (N66001-19-2-4031), NSF IIS-2044660, and ONR N00014-18-1-2826. We thank OpenAI for offering access to their API.

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

## A  More Details on Datasets

Table 6 shows more dataset statistics, and Table 7 shows the dataset citations and links from which we retrieved the datasets.

### A.1  Dataset-Specific Special Handling

For some datasets, we pre-process them into a unified multiple-choice or boolean format. We provide the details below.

**Com2Sense (paired).**  Com2Sense contains true and false statements that can be paired into complements. To utilize this pairing information, we place the two statements in each pair into the same statement group, and treat this as a multiple-choice dataset. Some statements in the dev set are not paired, so we discarded these examples.

**CycIC (mc).**  CycIC contains both multiple-choice and boolean QA problems. To keep consistency in evaluation, we use only the multiple-choice problems, which is the dominant problem type in this dataset.

**ComVE (task A).**  ComVE contains data for three tasks. Task A is assigning true/false labels to paired statements, similar to Com2Sense (paired). Task B and C are about choosing and generating explanations to a given statement being against commonsense. We use the data for task A.

**SKD (annotated).**  The annotated dataset of Symbolic Knowledge Distillation (SKD) contains LM-generated, semi-structured knowledge triples, where the head and tail events are connected by relations, such as

```
(PersonX doesn't like to wait, xIntent,
to get the job done).
```

Following West et al. (2021), we replace the name placeholders with random person names, and convert into natural language statements using templates adapted from Hwang et al. (2020). For example, the triple in the above example becomes

```
Arnold doesn't like to wait.  Because
Arnold wanted to get the job done.
```

We set the correctness label to be true iff the *valid* field has a positive value.

**I2D2 (annotated).**  The annotated dataset of I2D2 contains LM-generated commonsense statements with human-annotated correctness labels. We use the combination of annotated data in "Iter0" and "Iter2", because the data of "Iter1" is missing from the website.

### A.2  Conversion to Declarative Statements

From QA datasets, we create declarative statements from QA problems using the following method:

- If the problem contains a question, we convert the question and choice into a declarative statement using the question conversion model created by Chen et al. (2021).

- If the question is cloze-style, we replace the blank with the choice.

- If the question is an incomplete sentence and the choice is a continuation to it, we concatenate the question and the choice.

- If there is no question and the problem only asks to choose between some choices, we use the choice as the declarative statement.

- For boolean problems, we always use *yes* as the choice and create a single declarative statement for each problem. We use the original label as the correctness label of this statement.

## B  More Details on Method

### B.1  Training Objectives

**Binary classification loss.**  We defined the binary classification loss as

$$\mathcal{L}_{\text{bin}}(x_i, y_i) = \\ - y_i \log s(x_i) - (1 - y_i) \log(1 - s(x_i)).$$

To account for the fact that there are usually more incorrect statements than correct ones in the data produced from multiple-choice datasets, we divide this loss by the number of statements with the same correctness label in the same statement group. Therefore, the binary classification loss for the whole batch is

$$\mathcal{L}_{\text{bin}} = \\ \frac{1}{B_G} \sum_{j=1}^{B_G} \sum_{y \in \{0,1\}} \frac{\sum_{c=1}^{C_j} \mathbb{I}[y_{jc} = y] \mathcal{L}_{\text{bin}}(x_{jc}, y_{jc})}{\sum_{c=1}^{C_j} \mathbb{I}[y_{jc} = y]},$$

where $C_j$ is the number of statements in statement group $X_j$, $x_{jc}$ is the $c$th statement in $X_j$, and $\mathbb{I}$ is the indicator function.

**Multi-class loss.** We defined the multi-class loss as

$$\mathcal{L}_{\mathrm{mc}}(X_j) = -\log \frac{\exp z(x_{j*})}{\sum_{c=1}^{C_j} \exp z(x_{jc})}.$$

The multi-class loss for the whole batch is

$$\mathcal{L}_{\mathrm{mc}} = \frac{1}{B_G} \sum_{j=1}^{B_G} \mathcal{L}_{\mathrm{mc}}(X_j).$$

**Supervised contrastive loss.** We defined the supervised contrastive loss as

$$\mathcal{L}_{\mathrm{ctr}}(x_i, y_i) = -\log \frac{\sum_{k \in \mathcal{P}(i)} e^{\frac{\cos(\mathbf{h}(x_i), \mathbf{h}(x_k))}{\tau}}}{\sum_{k \in \mathcal{P}(i) \cup \mathcal{N}(i)} e^{\frac{\cos(\mathbf{h}(x_i), \mathbf{h}(x_k))}{\tau}}}.$$

The supervised contrastive loss for the whole batch is

$$\mathcal{L}_{\mathrm{ctr}} = \frac{1}{B_S} \sum_{i=1}^{B_S} \mathcal{L}_{\mathrm{ctr}}(x_i, y_i).$$

### B.2 Calibration

Our calibration is a post-hoc strategy and does not affect the task performance metrics we report in §5. This is because applying our calibration method – temperature scaling – does not affect the relative order of plausibility scores assigned to a given set of statements:

- For tasks with multiple-choice questions (§5.1), calibration does not affect the argmax prediction for the above reason.

- For commonsense knowledge filtering (§5.2), calibration does not affect the TPR/FPR numbers at each corresponding decision point, again for the above reason, so the ROC curves are valid.

- For True/False judgment problems (§5.1 and §5.3), calibration does not move the plausibility scores across the decision boundary. We use logit $z = 0.0$ (or equivalently, plausibility score $s = 0.5$) as the True/False boundary. A positive (or negative) logit remains positive (or negative) after applying the temperature.

## C More Details on Experimental Setup

Table 8 shows the hyperparamter settings for training VERA. These values are obtained from some moderate hyperparameter tuning, and we did not do extensive search due to training cost.

For tokenization, the T5 tokenizer tokenizes input so that it ends with the EOS token `` (token ID = 1). We manually configured the LLaMA tokenizer so that its output ends with the EOS token `` (token ID = 2), and does not contain the BOS token `` (token ID = 1). Models are trained for $S = 50k$ steps with $B_G = 64$ statement groups per batch, using the Adam optimizer (Kingma and Ba, 2014) with learning rate $\eta = 1 \times 10^{-5}$ for T5 encoder and $\eta = 2 \times 10^{-6}$ for LLaMA. We train models with the Huggingface Transformers and Accelerate libraries (Wolf et al., 2019; Gugger et al., 2022). For memory efficiency, during training, each statement is truncated to 128 tokens (which can accommodate more than 99% of the statements; see Table 6) and each statement group is capped to four statements.

### C.1 Definition of Metrics

**Multiple-choice accuracy.** For multiple-choice benchmarks, we report the multiple-choice accuracy:

$$Acc_{\mathrm{mc}} = \frac{1}{|\mathcal{D}|} \sum_{X_j \in \mathcal{D}} \mathbb{I}[x_{j*} = \arg\max_{x_{jc} \in X_j} s(x_{jc})].$$

**Boolean accuracy.** The boolean accuracy is defined as

$$Acc_{\mathrm{bool}} = \frac{1}{|\mathcal{D}|} \sum_{(x_i, y_i) \in \mathcal{D}} \mathbb{I}[y_i = \mathbb{I}[z(x_i) > 0]].$$

Boolean accuracy is applicable to balanced boolean benchmarks where there are roughly equal true and false statements (e.g., CommonsenseQA 2.0, Spatial Commonsense, StrategyQA, CREAK). Generally it is not a good metric for multiple-choice benchmarks and unbalanced boolean benchmarks.

**AUROC and AP.** For unbalanced boolean benchmarks (e.g., LM-generated knowledge filtering datasets), accuracy may not faithfully capture the model's performance. Instead, the metrics we use are the area under the ROC curve (AUROC) and the average precision (AP) for selecting the True statements. Statements are ranked based on their assigned raw scores, so that different score thresholds can be selected to construct the ROC and Precision-Recall curves. Aside from unbalanced boolean benchmarks, AUROC and AP are also applicable

to multiple-choice and balanced boolean benchmarks.

**Calibration.** To measure how well the verifier-predicted score reflects its confidence, we measure the ECE (Naeini et al., 2015) on the boolean benchmarks. ECE is computed as

$$\text{ECE} = \sum_{m=1}^{M} \frac{|B_m|}{|\mathcal{D}|} \cdot \Big| \text{Acc}(B_m) - \text{Score}(B_m) \Big|$$

$$= \sum_{m=1}^{M} \frac{|B_m|}{|\mathcal{D}|} \cdot \Big| \frac{1}{|B_m|} \sum_{(x_i, y_i) \in B_m} \mathbb{I}[y_i = 1]$$

$$- \frac{1}{|B_m|} \sum_{(x_i, y_i) \in B_m} s(x_i) \Big|, \quad (1)$$

where $M$ is the number of bins which bucket data points with similar predictions, and $B_m \subseteq \mathcal{D}$ is the subset of data points that fall into the $m$-th bin. We use $M = 10$ equal-sized bins when computing ECE.

## C.2 Details on Baseline Models

**SKD Critic.** West et al. (2021) trained a critic model that filters incorrect commonsense knowledge generated by their symbolic knowledge distillation (SKD) method. This critic model is based on RoBERTa-large (Liu et al., 2019) and is finetuned on $8k$ GPT-3-generated commonsense knowledge sentences with human-annotated true/false labels. The model predicts a $[0, 1]$ score $s$ which we use as the final score, and we let the logit $z = \sigma^{-1}(s)$.

**I2D2 Critic.** Bhagavatula et al. (2022) trained a critic model that filters incorrect commonsense knowledge generated by their I2D2 method. This critic model is based on RoBERTa-large (Liu et al., 2019) and is finetuned on $12k$ I2D2-generated commonsense knowledge sentences with human-annotated true/false labels. Given an input statement, the model predicts two logits: $t$ for the True label and $f$ for the False label. We let the logit $z = t - f$ and the score $s = \sigma(t - f)$. We use the critic model trained in the final iteration (i.e., "Iter 2" in I2D2[3]).

**UnifiedQA-v2.** UnifiedQA-v2 (Khashabi et al., 2022) is a general-purpose QA model trained on datasets with a variety of input formats, including boolean datasets. When the input is a declarative statement, the model is trained to output either "yes"

or "no". We use this feature of the model and make it act as a commonsense statement verifier. For an input statement, we compute the logits received by "yes" and "no" in the decoder, denoted as $t$ and $f$, respectively. We let the logit $z = t - f$ and the score $s = \sigma(t - f)$. We use the largest version of this model, UnifiedQA-v2-11b.[4]

**Entailer.** Entailer (Tafjord et al., 2022) is a model trained to construct proof trees for scientific commonsense hypotheses. This multi-angle model can be used in three ways: (1) given a hypothesis, generate a set of premises that may entail it; (2) given a hypothesis, predict a score that reflects the model's belief in it; (3) given a hypothesis and set of premises, predict a score that reflects whether there is a valid entailment between them. We use (2) as a commonsense statement verifier. The model predicts a $[0, 1]$ score $s$ which we use as the final score, and we let the logit $z = \sigma^{-1}(s)$. We use the largest version of this model, Entailer-11b.[5]

**GPT-3.5.** GPT-3.5 (OpenAI, 2022b) is a series of general-purpose autoregressive decoder-only LMs. To make it act as a commonsense verifier, we use the following input prompt:

```
Question: Based on commonsense knowledge, is the following

statement correct? Please answer yes or no.

Statement: {statement}

Answer:
```

We query the OpenAI Completions API[6] with this prompt and compute the logits received by " Yes" and " No" in the next-token prediction, denoted as $t$ and $f$, respectively. We let the logit $z = t - f$ and the score $s = \sigma(t - f)$. We experimented with several prompt formats and found the one presented above to have the best performance, and in most cases, " Yes" and " No" together receive most of the probability mass during next-token prediction. We also experimented with several models in the GPT-3 (Brown et al., 2020) and GPT-3.5 series, and found GPT-3.5 (text-davinci-002) to work the best.

Additionally, we report a baseline where the (negated) language modeling perplexity is used for

---

[3]https://gengen.apps.allenai.org/

[4]https://huggingface.co/allenai/unifiedqa-v2-t5-11b-1251000

[5]https://huggingface.co/allenai/entailer-11b

[6]https://platform.openai.com/docs/api-reference/completions

commonsense plausibility. Note that the plausibility scores derived this way are not normalized, and we only use them for ranking purposes. For this baseline, we use GPT-3.5 (`text-davinci-002`) as the base model, and name it as "PPL (GPT-3.5)".

**ChatGPT and GPT-4.** ChatGPT (OpenAI, 2022a) and GPT-4 (OpenAI, 2023) are optimized for chat. To make them act as a commonsense verifier, we use the same input prompt as for GPT-3.5, without the "`Answer:`" line. We query the OpenAI Chat API[7] with this prompt in a user message, and obtain the first token of the assistant message in the response. Besides this zero-shot setting, we additionally report a few-shot chain-of-thought (Wei et al., 2022) setting with 5 in-domain examples, formatted as additional user-assistant message pairs prior to the query user message.

Since the API does not provide token logits, we let the score $s = 1.0$ when this token is "Yes", and $s = 0.0$ when this token is "No". In the unlikely case that this token is neither, we let $s = 0.5$. We add a small random noise to the score. This is to arbitrate potentially multiple positive predictions within statement groups from multiple-choice QA problems, and to enable plotting the ROC and precision-recall curves. Note that this is not an ideal solution and may cause under-estimation of ChatGPT and GPT-4's performance.

**Flan-T5.** Flan-T5 (Chung et al., 2022) is a series of sequence-to-sequence LMs instruction-finetuned on massive number of tasks. To make it act as a commonsense verifier, we use the same input prompt as for GPT-3.5. We compute the logits received by "yes" and "no" in the first token prediction in the decoder, denoted as $t$ and $f$, respectively. We let the logit $z = t - f$ and the score $s = \sigma(t - f)$. We experimented with several prompt formats and found the one presented above to have the best performance, and in most cases, "yes" and "no" together receive most of the probability mass during the token prediction. We use the largest version of this model, `Flan-T5-XXL`.[8] Note that some unseen benchmarks are in the training data of Flan-T5; see Table 7 for details on data contamination.

## D More Evaluation Results

Figure 5 is an expansion of Table 2 and additionally shows the precision-recall curves on problem-solving benchmarks. Table 9, Table 10, and Table 11 show the per-dataset breakdown of the accuracy numbers in Figure 5. Figure 6 is an expansion of Figure 2 and additionally shows the precision-recall curves on knowledge-filtering benchmarks. Table 12 shows the per-dataset breakdown of the accuracy numbers in Table 3.

## E Further Analysis

**Format: Verification vs. QA.** In this paper, we have been using the verification format to approach problem-solving tasks. But do we lose something when compared to using the QA format? In Figure 7 we compare how well existing models can solve problems in the verification format and the QA format. Verification format does fall behind QA format, especially with models trained exclusively in QA format (i.e., UnifiedQA-v2). We also trained a sequence-to-sequence model in QA format on the same multiple-choice data as VERA. It leads VERA by 1.5% on seen multiple-choice benchmarks. We hypothesize that this is because verification models only see one option at a time, whereas QA models can see all choices of a problem at the same time and thus can do comparative ranking.

In addition to the performance loss, a verification model does lose the generative capability possessed by some QA models that are generative (e.g., UnifiedQA in Khashabi et al. (2020)), and it has to run $C$ times to solve a $C$-way multiple-choice problem, whereas QA models (e.g., UnifiedQA in Khashabi et al. (2020), Unicorn in Lourie et al. (2021)) need to run only once.

However, verification models can perform some tasks that generative QA models cannot cover. They can classify the correctness of declarative statements, without having to convert them into questions in the first place. They can also reflect on the answer produced by a generative QA model, and provide a level of confidence. We argue that verification models and generative QA models have different best-application scenarios and are sometimes complementary to each other.

---

[7]https://platform.openai.com/docs/api-reference/chat

[8]https://huggingface.co/google/flan-t5-xxl

| Abbr. | Name | Domain | Format | # Train Ex. | Aug | # Dev Ex. | # Statements | # True | # False |
|---|---|---|---|---|---|---|---|---|---|
| | | | STAGE A TRAINING | | | | | | |
| Atomic2020 | Atomic2020 | | multiple-choice (4) | 803541 | | 70731 | 282924 | 70731 | 212193 |
| GenericsKB | GenericsKB | | multiple-choice (4) | 775820 | | 96977 | 387908 | 96977 | 290931 |
| **Total** | | | | **1579361** | | **167708** | **670832** | **167708** | **503124** |
| | | | STAGE B TRAINING (SEEN) | | | | | | |
| OBQA | OpenBookQA | scientific | multiple-choice (4) | 4957 | ✓ | 500 | 2000 | 500 | 1500 |
| ARC_e | ARC (easy) | scientific | multiple-choice (4) | 2251 | ✓ | 570 | 2281 | 570 | 1711 |
| ARC_h | ARC (hard) | scientific | multiple-choice (4) | 1119 | ✓ | 299 | 1194 | 299 | 895 |
| AI2Sci_e | AI2 Science (elem) | scientific | multiple-choice (4) | 623 | ✓ | 123 | 489 | 123 | 366 |
| AI2Sci_m | AI2 Science (middle) | scientific | multiple-choice (4) | 605 | ✓ | 125 | 502 | 125 | 377 |
| CSQA | CommonsenseQA | general | multiple-choice (5) | 9741 | ✓ | 1221 | 6099 | 1221 | 4878 |
| QASC | QASC | scientific | multiple-choice (8) | 8134 | ✓ | 926 | 7408 | 926 | 6482 |
| PIQA | Physical IQA | physical | multiple-choice (2) | 16113 | | 1838 | 3676 | 1838 | 1838 |
| SIQA | Social IQA | social | multiple-choice (3) | 33410 | ✓ | 1954 | 5861 | 1954 | 3907 |
| WG | Winogrande | general | multiple-choice (2) | 40398 | | 1267 | 2534 | 1267 | 1267 |
| C2S | Com2Sense (paired) | general | multiple-choice (2) | 804 | | 391 | 782 | 391 | 391 |
| SciQ | SciQ | scientific | multiple-choice (4) | 11679 | ✓ | 1000 | 4000 | 1000 | 3000 |
| QuaRel | QuaRel | qualitative | multiple-choice (2) | 1941 | | 278 | 556 | 278 | 278 |
| QuaRTz | QuaRTz | qualitative | multiple-choice (2) | 2696 | | 384 | 768 | 384 | 384 |
| CycIC | CycIC (mc) | general | multiple-choice (5) | 6521 | | 907 | 4535 | 907 | 3628 |
| ComVE | ComVE (task A) | general | multiple-choice (2) | 10000 | | 997 | 1994 | 997 | 997 |
| CSQA2 | CommonsenseQA 2.0 | general | boolean | 9264 | | 2541 | 2541 | 1225 | 1316 |
| SKD_anno | SKD (annotated) | | boolean | 7980 | | 1015 | 1015 | 803 | 212 |
| I2D2_anno | I2D2 (annotated) | | boolean | 26206 | | 13094 | 13094 | 6158 | 6936 |
| **Total** | | | | **194442** | | **29430** | **61329** | **20966** | **40363** |
| | | | EVALUATION (UNSEEN TYPE 1) | | | | | | |
| WSC | WSC | general | multiple-choice (2) | 0 | | 273 | 546 | 273 | 273 |
| COPA | COPA | general | multiple-choice (2) | 0 | | 500 | 1000 | 500 | 500 |
| NumerSense | NumerSense | quantitative | multiple-choice (11) | 0 | | 200 | 2200 | 200 | 2000 |
| PROST | PROST | physical | multiple-choice (4) | 0 | | 18736 | 74944 | 18736 | 56208 |
| SpatialCS | Spatial Commonsense | physical | boolean | 0 | | 1448 | 1448 | 724 | 724 |
| Rainier_anno | Rainier (annotated) | | boolean | 0 | | 591 | 591 | 424 | 167 |
| **Total** | | | | **0** | | **21748** | **80729** | **20857** | **59872** |
| | | | EVALUATION (UNSEEN TYPE 2) | | | | | | |
| SWAG | SWAG | | multiple-choice (4) | 0 | | 20006 | 80024 | 20006 | 60018 |
| HellaSwag | HellaSwag | | multiple-choice (4) | 0 | | 10042 | 40168 | 10042 | 30126 |
| CODAH | CODAH | | multiple-choice (4) | 0 | | 2776 | 11104 | 2776 | 8328 |
| SCT | Story Cloze Test | | multiple-choice (2) | 0 | | 1871 | 3742 | 1871 | 1871 |
| $\alpha$NLI | $\alpha$NLI | | multiple-choice (2) | 0 | | 1532 | 3064 | 1532 | 1532 |
| StrategyQA | StrategyQA | | boolean | 0 | | 229 | 229 | 107 | 122 |
| CREAK | CREAK | | boolean | 0 | | 1371 | 1371 | 691 | 680 |
| **Total** | | | | **0** | | **37827** | **139702** | **37025** | **102677** |

Table 5: Datasets and statistics. Data sourced from commonsense KBs are listed under STAGE A TRAINING, and data sourced from commonsense QA datasets are listed under STAGE B TRAINING. The number in parentheses under the **Format** column represents the number of choices per question. The **Aug** column indicates whether LM-augmented falsehoods are generated for each dataset. The last three columns are the number of total, correct and incorrect statements in the development set. See Table 6 for more dataset statistics, and Table 7 for full citations and sources for these datasets.

| Abbr. | Name | Statement Length | | | | | |
|---|---|---|---|---|---|---|---|
| | | min | median | 90% | 95% | 99% | max |
| STAGE A TRAINING | | | | | | | |
| Atomic2020 | Atomic2020 | 5 | 19 | 24 | 26 | 30 | 57 |
| GenericsKB | GenericsKB | 4 | 13 | 22 | 24 | 28 | 82 |
| STAGE B TRAINING (SEEN) | | | | | | | |
| OBQA | OpenBookQA | 5 | 16 | 29 | 36 | 56 | 74 |
| ARC_e | ARC (easy) | 6 | 24 | 50 | 60 | 86 | 111 |
| ARC_h | ARC (hard) | 7 | 30 | 59 | 70 | 94 | 138 |
| AI2Sci_e | AI2 Science (elem) | 7 | 29 | 63 | 79 | 455 | 473 |
| AI2Sci_m | AI2 Science (middle) | 7 | 24 | 58 | 72 | 511 | 536 |
| CSQA | CommonsenseQA | 5 | 18 | 28 | 32 | 43 | 73 |
| QASC | QASC | 5 | 13 | 19 | 21 | 24 | 30 |
| PIQA | Physical IQA | 5 | 26 | 62 | 80 | 120 | 256 |
| SIQA | Social IQA | 10 | 28 | 38 | 41 | 51 | 70 |
| WG | Winogrande | 17 | 24 | 31 | 34 | 38 | 42 |
| C2S | Com2Sense (paired) | 12 | 24 | 34 | 38 | 44 | 55 |
| SciQ | SciQ | 6 | 19 | 29 | 34 | 48 | 75 |
| QuaRel | QuaRel | 15 | 39 | 62 | 80 | 101 | 107 |
| QuaRTz | QuaRTz | 9 | 29 | 44 | 49 | 73 | 78 |
| CycIC | CycIC (mc) | 6 | 31 | 59 | 67 | 97 | 122 |
| ComVE | ComVE (task A) | 4 | 10 | 14 | 16 | 20 | 28 |
| CSQA2 | CommonsenseQA 2.0 | 5 | 14 | 24 | 29 | 38 | 58 |
| SKD_anno | SKD (annotated) | 13 | 20 | 25 | 27 | 31 | 37 |
| I2D2_anno | I2D2 (annotated) | 5 | 15 | 21 | 24 | 31 | 41 |
| EVALUATION (UNSEEN TYPE 1) | | | | | | | |
| WSC | WSC | 10 | 22 | 33 | 39 | 45 | 48 |
| COPA | COPA | 10 | 17 | 21 | 23 | 26 | 28 |
| NumerSense | NumerSense | 6 | 13 | 20 | 23 | 32 | 36 |
| PROST | PROST | 17 | 42 | 63 | 68 | 78 | 78 |
| SpatialCS | Spatial Commonsense | 9 | 12 | 17 | 18 | 19 | 20 |
| Rainier_anno | Rainier (annotated) | 5 | 12 | 19 | 21 | 29 | 33 |
| EVALUATION (UNSEEN TYPE 2) | | | | | | | |
| SWAG | SWAG | 12 | 30 | 46 | 52 | 67 | 148 |
| HellaSwag | HellaSwag | 15 | 103 | 134 | 140 | 149 | 181 |
| CODAH | CODAH | 5 | 21 | 31 | 34 | 45 | 73 |
| SCT | Story Cloze Test | 29 | 56 | 69 | 72 | 78 | 89 |
| $\alpha$NLI | $\alpha$NLI | 17 | 35 | 44 | 47 | 53 | 65 |
| StrategyQA | StrategyQA | 6 | 14 | 20 | 22 | 25 | 30 |
| CREAK | CREAK | 8 | 14 | 20 | 22 | 29 | 50 |

Table 6: More dataset statistics. This table shows the percentiles of statement lengths (as in number of T5 tokens) in each dataset.

| Abbr. | Name | Citation | Link | In Flan-T5? |
|---|---|---|---|---|
| | | STAGE A TRAINING | | |
| Atomic2020 | Atomic2020 | Hwang et al. (2020) | https://allenai.org/data/atomic-2020 | yes |
| GenericsKB | GenericsKB | Bhakthavatsalam et al. (2020) | https://allenai.org/data/genericskb | yes |
| | | STAGE B TRAINING (SEEN) | | |
| OBQA | OpenBookQA | Mihaylov et al. (2018) | https://github.com/allenai/unifiedqa | yes |
| ARC_e | ARC (easy) | Clark et al. (2018) | https://github.com/allenai/unifiedqa | yes |
| ARC_h | ARC (hard) | Clark et al. (2018) | https://github.com/allenai/unifiedqa | yes |
| AI2Sci_e | AI2 Science (elem) | Clark et al. (2018) | https://github.com/allenai/unifiedqa | no |
| AI2Sci_m | AI2 Science (middle) | Clark et al. (2018) | https://github.com/allenai/unifiedqa | no |
| CSQA | CommonsenseQA | Talmor et al. (2019) | https://github.com/allenai/unifiedqa | yes |
| QASC | QASC | Khot et al. (2019) | https://github.com/allenai/unifiedqa | yes |
| PIQA | Physical IQA | Bisk et al. (2019) | https://github.com/allenai/unifiedqa | yes |
| SIQA | Social IQA | Sap et al. (2019) | https://github.com/allenai/unifiedqa | yes |
| WG | Winogrande | Sakaguchi et al. (2019) | https://github.com/allenai/unifiedqa | yes |
| C2S | Com2Sense (paired) | Singh et al. (2021) | https://github.com/PlusLabNLP/Com2Sense/tree/master/data | yes |
| SciQ | SciQ | Welbl et al. (2017) | https://allenai.org/data/sciq | yes |
| QuaRel | QuaRel | Tafjord et al. (2018) | https://allenai.org/data/quarel | yes |
| QuaRTz | QuaRTz | Tafjord et al. (2019) | https://allenai.org/data/quartz | yes |
| CycIC | CycIC (mc) | – | https://leaderboard.allenai.org/cycic/submissions/get-started | no |
| ComVE | ComVE (task A) | Wang et al. (2020) | https://github.com/wangcunxiang/SemEval2020-Task4-Commonsense-Validation-and-Explanation | no |
| CSQA2 | CommonsenseQA 2.0 | Talmor et al. (2021) | https://github.com/allenai/csqa2/tree/master/dataset | no |
| SKD_anno | SKD (annotated) | West et al. (2021) | https://github.com/peterwestai2/symbolic-knowledge-distillation/tree/main/purification_code | no |
| I2D2_anno | I2D2 (annotated) | Bhagavatula et al. (2022) | https://gengen.apps.allenai.org | no |
| | | EVALUATION (UNSEEN TYPE 1) | | |
| WSC | WSC | Levesque et al. (2011) | https://huggingface.co/datasets/winograd_wsc | yes |
| COPA | COPA | Gordon et al. (2011) | https://huggingface.co/datasets/super_glue | yes |
| NumerSense | NumerSense | Lin et al. (2020) | https://github.com/INK-USC/NumerSense/tree/main/data | yes |
| PROST | PROST | Aroca-Ouellette et al. (2021) | https://huggingface.co/datasets/corypaik/prost | yes |
| SpatialCS | Spatial Commonsense | Liu et al. (2022c) | https://github.com/xxxiaol/spatial-commonsense | no |
| Rainier_anno | Rainier (annotated) | Liu et al. (2022b) | https://github.com/liujch1998/rainier | no |
| | | EVALUATION (UNSEEN TYPE 2) | | |
| SWAG | SWAG | Zellers et al. (2018) | https://github.com/rowanz/swagaf/tree/master/data | yes |
| HellaSwag | HellaSwag | Zellers et al. (2019) | https://github.com/rowanz/hellaswag/tree/master/data | yes |
| CODAH | CODAH | Chen et al. (2019) | https://github.com/Websail-NU/CODAH/tree/master/data | yes |
| SCT | Story Cloze Test | Mostafazadeh et al. (2016) | https://cs.rochester.edu/nlp/rocstories/ | yes |
| $\alpha$NLI | $\alpha$NLI | Bhagavatula et al. (2019) | https://leaderboard.allenai.org/anli/submissions/get-started | yes |
| StrategyQA | StrategyQA | Geva et al. (2021) | https://github.com/eladsegal/strategyqa/tree/main/data/strategyqa | yes |
| CREAK | CREAK | Onoe et al. (2021) | https://github.com/yasumasaonoe/creak/tree/main/data/creak | yes |

Table 7: More dataset details. We show the link from which we retrieved each dataset, and whether each dataset is included in the training data of Flan-T5.

| Symbol | Value | Description |
|---|---|---|
| $L$ | 128 | Max number of tokens in the input statement |
| $B_G$ | 64 | Number of statement groups per batch |
| $C$ | 4 | Max number of statements in each group, during training |
| $B_S$ | 256 | Max number of statements per batch, during training |
| $S$ | 50,000 | Total number of training steps in each stage |
| $\eta_{\text{T5}}$ | $1 \times 10^{-5}$ | Learning rate for VERA with T5 encoder backbone |
| $\eta_{\text{LLaMA}}$ | $2 \times 10^{-6}$ | Learning rate for VERA with LLaMA backbone |
| $\alpha$ | 1.0 | Weight of binary classification loss |
| $\beta$ | 1.0 | Weight of multiple-choice loss |
| $\gamma$ | 0.1 | Weight of supervised contrastive loss |
| $\tau$ | 0.05 | Temperature in supervised contrastive loss |

Table 8: Hyperparameter settings.

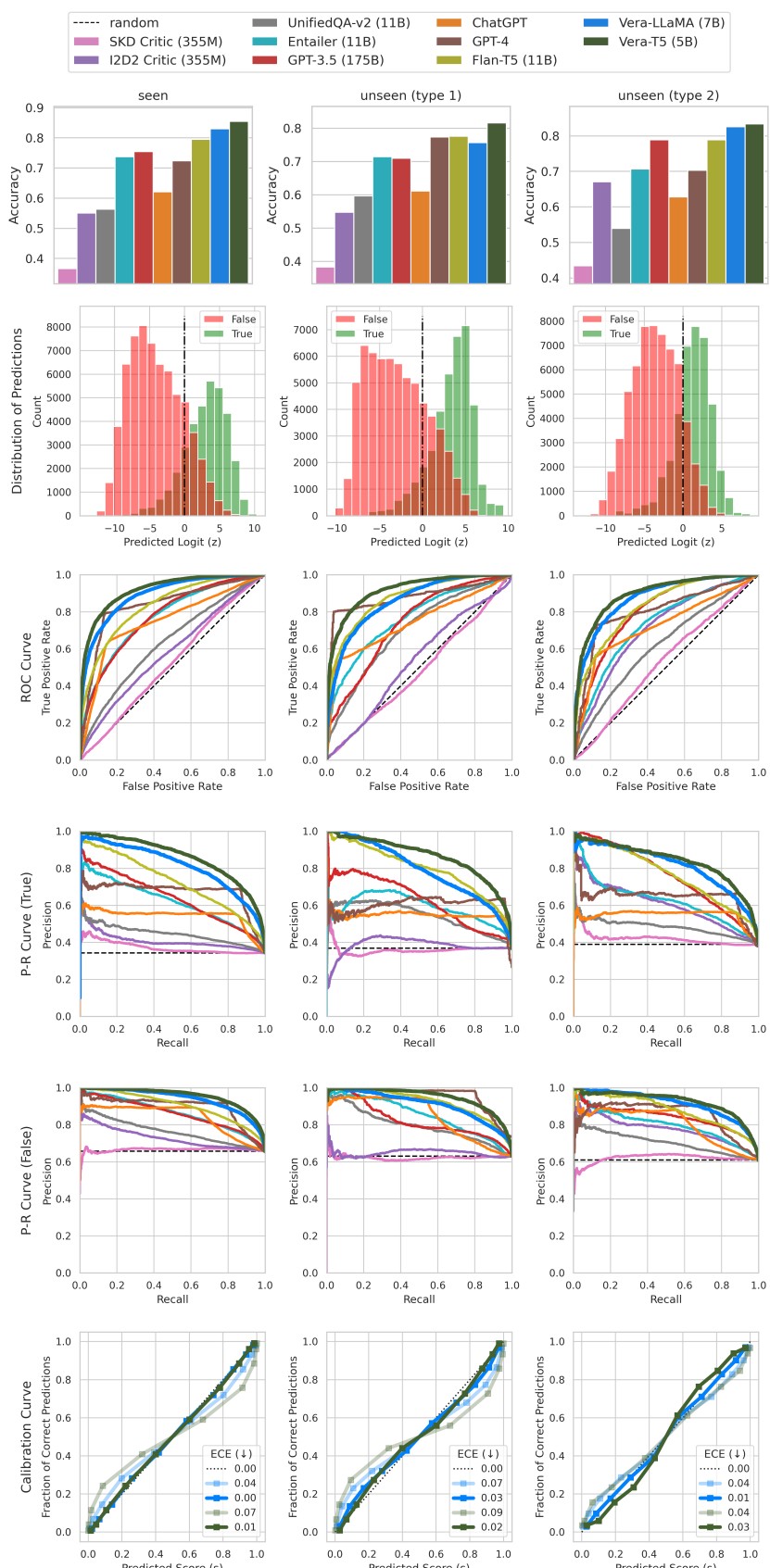

Figure 5: Results on problem-solving with VERA on seen and unseen benchmarks. Average results on the development sets are reported. Accuracy across different parts (seen, unseen (type 1), unseen (type 2)) are not directly comparable due to different underlying benchmarks. For calibration curves, curves with saturated colors are results after applying post hoc calibration (§3.3), while curves with faded colors are results from the raw logits.

| Dataset → | All | MC | Bool | OBQA | ARC_e | ARC_h | AI2Sci_e | AI2Sci_m | CSQA | QASC | PIQA | SIQA | WG | C2S | SciQ | QuaRel | QuaRTz | CycIC | ComVE | CSQA2 |
|---|---|---|---|---|---|---|---|---|---|---|---|---|---|---|---|---|---|---|---|---|
| SKD Critic (355M) | 36.64 | 35.96 | 47.60 | 27.60 | 29.12 | 23.08 | 25.20 | 28.00 | 20.15 | 12.42 | 53.86 | 39.20 | 50.28 | 51.41 | 27.30 | 53.60 | 55.73 | 25.80 | 52.56 | 47.60 |
| I2D2 Critic (355M) | 55.03 | 55.75 | 43.65 | 44.80 | 55.61 | 35.79 | 55.28 | 50.40 | 61.51 | 45.25 | 67.36 | 56.45 | 55.56 | 63.17 | 55.10 | 61.51 | 60.16 | 35.72 | 88.26 | 43.65 |
| UnifiedQA-v2 (11B) | 56.33 | 56.34 | 56.05 | 54.60 | 48.77 | 39.46 | 48.78 | 43.20 | 44.23 | 32.61 | 63.98 | 52.10 | 70.64 | 75.96 | 42.20 | 81.65 | 71.35 | 48.29 | 83.65 | 56.05 |
| Entailer (11B) | 73.79 | 74.90 | 56.00 | 74.40 | 81.93 | 64.88 | 77.24 | 82.40 | 67.81 | 57.56 | 78.78 | 64.33 | 77.11 | 82.86 | 76.90 | 85.97 | 76.56 | 52.81 | 96.89 | 56.00 |
| PPL (GPT-3.5) | 66.02 | 66.02 | – | 45.20 | 70.12 | 45.15 | 69.92 | 62.70 | 61.92 | 57.02 | 81.55 | 51.23 | 71.37 | 72.56 | 86.80 | 71.22 | 70.91 | 53.31 | 85.37 | – |
| GPT-3.5 (175B) | 75.41 | 76.34 | 60.55 | 74.20 | 85.79 | 68.90 | 84.55 | 80.80 | 66.91 | 62.85 | 84.17 | 65.30 | 72.53 | 81.33 | 86.00 | 83.09 | 76.04 | 51.60 | 97.39 | 60.55 |
| ChatGPT | 62.11 | 61.52 | 71.65 | 60.80 | 65.44 | 57.19 | 63.41 | 68.00 | 39.64 | 42.01 | 67.36 | 52.20 | 61.33 | 76.73 | 60.70 | 74.10 | 72.66 | 29.66 | 93.08 | 71.65 |
| + 5-shot CoT | 65.19 | 65.19 | – | 62.40 | 77.33 | 62.88 | 72.36 | 69.84 | 46.52 | 47.52 | 68.59 | 52.25 | 59.98 | 82.14 | 69.77 | 65.83 | 72.14 | 42.56 | 90.98 | – |
| GPT-4 | 72.35 | 71.81 | 81.00 | 76.00 | 69.00 | 72.00 | 80.00 | 80.00 | 43.00 | 44.00 | 73.00 | 57.00 | 77.00 | 94.00 | 70.00 | 86.00 | 80.00 | 53.00 | 95.00 | 81.00 |
| + 5-shot CoT | 74.96 | 74.96 | – | 79.80 | 79.00 | 70.00 | 87.00 | 89.11 | 31.31 | 44.00 | 80.00 | 67.00 | 82.18 | 91.92 | 80.00 | 82.00 | 87.00 | 57.00 | 92.08 | – |
| Flan-T5 (11B) | 79.50 | 80.58 | 62.25 | 79.60 | 85.79 | 71.24 | 86.99 | 81.60 | 69.21 | 64.58 | 83.95 | 73.23 | 84.69 | 84.40 | 80.80 | 92.81 | 82.03 | 69.90 | 98.40 | 62.25 |
| VERA-LLaMA (7B) | 82.99 | 84.18 | 63.85 | 80.20 | 84.39 | 75.92 | 88.62 | 82.40 | 76.17 | 71.38 | 85.91 | 79.89 | 87.92 | 83.63 | 90.00 | 92.09 | 80.99 | 89.42 | 97.99 | 63.85 |
| VERA-T5 (5B) | 85.51 | 86.57 | 68.60 | 83.20 | 88.07 | 78.60 | 93.50 | 86.40 | 77.97 | 73.33 | 88.47 | 80.14 | 92.42 | 85.93 | 88.80 | 93.88 | 84.90 | 91.73 | 97.79 | 68.60 |

Table 9: Results on **seen** benchmarks. Accuracy on the development set is reported.

| Dataset → | All | MC | Bool | WSC | COPA | NumerSense | PROST | SpatialCS |
|---|---|---|---|---|---|---|---|---|
| SKD Critic (355M) | 38.34 | 35.83 | 48.41 | 54.21 | 53.00 | 11.50 | 24.60 | 48.41 |
| I2D2 Critic (355M) | 54.79 | 54.43 | 56.22 | 80.59 | 72.80 | 35.00 | 29.35 | 56.22 |
| UnifiedQA-v2 (11B) | 59.73 | 55.10 | 78.25 | 71.79 | 81.20 | 35.00 | 32.40 | 78.25 |
| Entailer (11B) | 71.47 | 68.05 | 85.15 | 86.08 | 92.40 | 51.00 | 42.70 | 85.15 |
| GPT-3.5 (175B) | 71.03 | 70.73 | 72.24 | 85.71 | 87.00 | 66.50 | 43.70 | 72.24 |
| ChatGPT | 61.20 | 54.69 | 87.22 | 73.26 | 58.80 | 47.50 | 39.20 | 87.22 |
| GPT-4 | 77.40 | 71.75 | 100.00 | 85.00 | 64.00 | 69.00 | 69.00 | 100.00 |
| Flan-T5 (11B) | 77.62 | 73.22 | 95.23 | 90.48 | 93.00 | 57.50 | 51.90 | 95.23 |
| VERA-LLaMA (7B) | 75.71 | 74.06 | 82.32 | 94.14 | 91.80 | 65.00 | 45.30 | 82.32 |
| VERA-T5 (5B) | 81.65 | 78.70 | 93.44 | 94.51 | 93.40 | 66.50 | 60.40 | 93.44 |

Table 10: Results on **unseen (type 1)** benchmarks. Accuracy on the development set is reported.

| Dataset → | All | MC | Bool | SWAG | HellaSwag | CODAH | SCT | αNLI | StrategyQA | CREAK |
|---|---|---|---|---|---|---|---|---|---|---|
| SKD Critic (355M) | 43.40 | 40.11 | 51.62 | 26.95 | 30.45 | 29.35 | 62.75 | 51.04 | 50.66 | 52.59 |
| I2D2 Critic (355M) | 67.11 | 70.42 | 58.84 | 72.15 | 53.30 | 67.30 | 88.24 | 71.08 | 52.40 | 65.28 |
| UnifiedQA-v2 (11B) | 53.95 | 52.83 | 56.77 | 31.75 | 36.60 | 49.00 | 82.04 | 64.75 | 49.34 | 64.19 |
| Entailer (11B) | 70.72 | 70.63 | 70.94 | 52.45 | 47.65 | 80.70 | 94.39 | 77.94 | 60.26 | 81.62 |
| GPT-3.5 (175B) | 78.87 | 80.21 | 75.53 | 73.40 | 70.40 | 85.05 | 95.56 | 76.63 | 62.88 | 88.18 |
| ChatGPT | 62.83 | 56.21 | 79.39 | 43.70 | 42.95 | 56.75 | 77.34 | 60.31 | 67.69 | 91.10 |
| GPT-4 | 70.29 | 66.20 | 80.50 | 57.00 | 40.00 | 66.00 | 93.00 | 75.00 | 70.00 | 91.00 |
| Flan-T5 (11B) | 78.89 | 80.52 | 74.81 | 69.20 | 64.55 | 89.60 | 98.45 | 80.81 | 61.14 | 88.48 |
| VERA-LLaMA (7B) | 82.56 | 86.11 | 73.71 | 79.30 | 83.90 | 88.95 | 98.61 | 79.77 | 62.45 | 84.97 |
| VERA-T5 (5B) | 83.37 | 86.66 | 75.13 | 76.30 | 85.90 | 88.60 | 98.56 | 83.94 | 65.07 | 85.19 |

Table 11: Results on **unseen (type 2)** benchmarks. Accuracy on the development set is reported.

| Generator | Filter | QA | Avg | OBQA | ARC_e | ARC_h | AI2Sci_e | AI2Sci_m | CSQA | QASC | PIQA | SIQA | WG |
|---|---|---|---|---|---|---|---|---|---|---|---|---|---|
| – | – | UnifiedQA-large | 60.45 | 70.20 | 69.12 | 55.85 | 69.11 | 64.80 | 61.43 | 43.09 | 63.66 | 53.84 | 53.35 |
| GPT-3 (davinci) | – | UnifiedQA-large | 67.44 | 74.60 | 75.44 | 64.55 | 69.92 | 72.80 | 70.19 | 63.82 | 67.74 | 58.70 | 56.59 |
| GPT-3 (davinci) | VERA | UnifiedQA-large | **70.67** | **77.60** | **80.00** | **67.56** | **78.05** | **78.40** | **71.91** | **66.20** | **70.35** | **59.37** | **57.22** |

| Generator | Filter | QA | Avg | OBQA | ARC_e | ARC_h | AI2Sci_e | AI2Sci_m | CSQA | QASC | PIQA | SIQA | WG |
|---|---|---|---|---|---|---|---|---|---|---|---|---|---|
| – | – | UnifiedQA-large | 60.45 | 70.20 | 69.12 | 55.85 | 69.11 | 64.80 | 61.43 | 43.09 | 63.66 | 53.84 | 53.35 |
| Rainier-large | – | UnifiedQA-large | 61.78 | 69.40 | 66.84 | 52.84 | 68.29 | 57.60 | **68.30** | 54.86 | 65.51 | 56.81 | 57.38 |
| Rainier-large | VERA | UnifiedQA-large | **64.88** | **73.40** | **71.05** | **57.19** | **73.98** | **67.20** | **68.30** | **55.51** | **67.52** | **56.96** | **57.70** |

Table 12: Results of introducing VERA into the Generated Knowledge Prompting pipeline (Liu et al., 2021). Accuracy on the development set is reported.

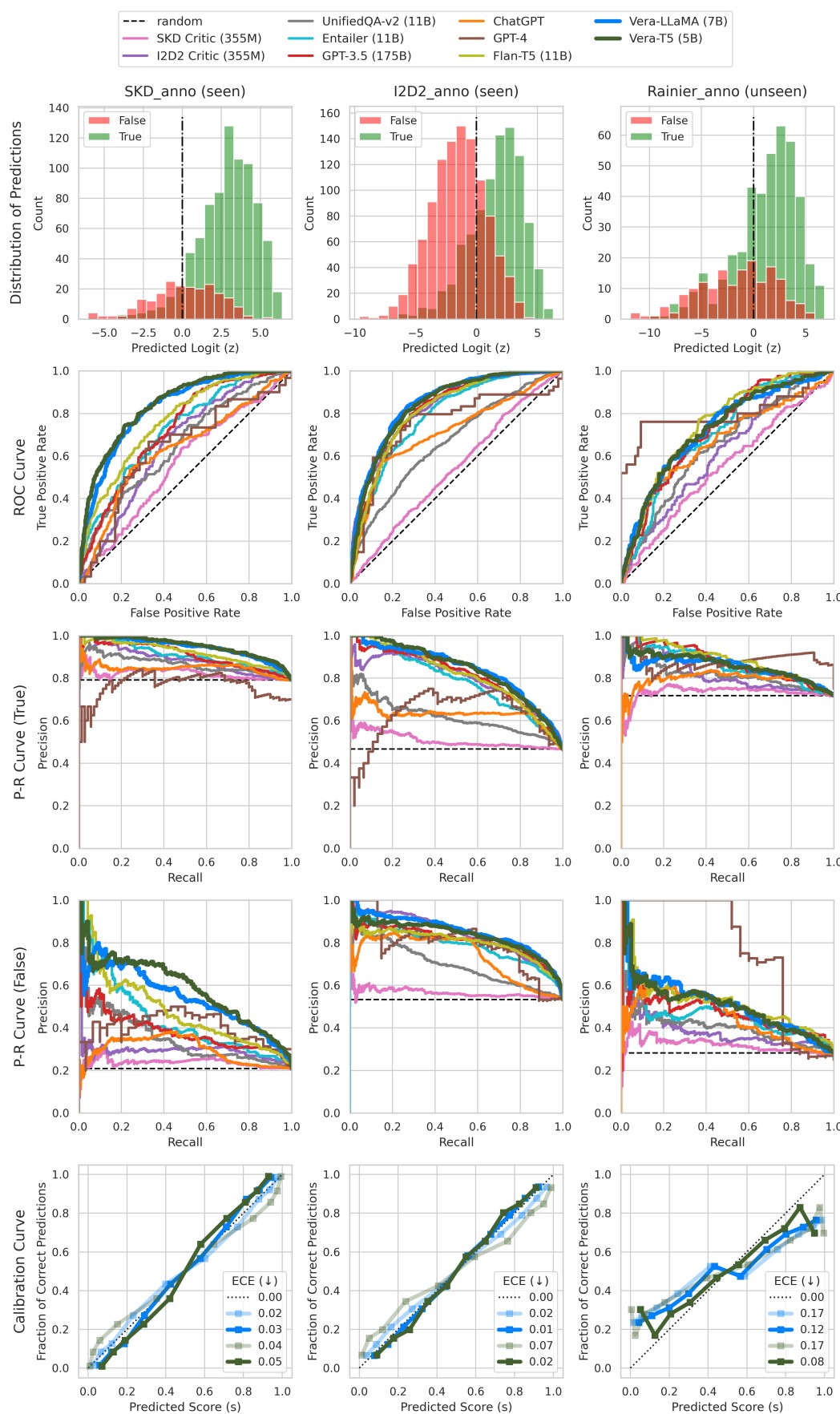

Figure 6: Results for filtering LM-generated commonsense knowledge with VERA. Results on the development sets are reported.

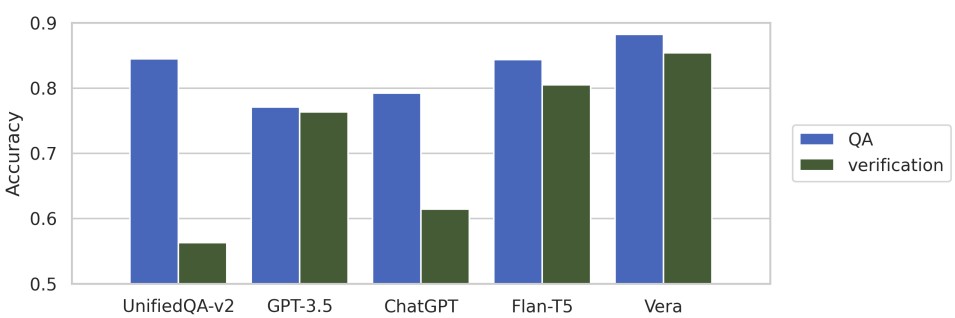

Figure 7: Comparing verification and QA, the two different formats for problem-solving tasks. Average accuracy on the development sets of the seen multiple-choice benchmarks is reported. We use `text-davinci-002` as GPT-3.5 here, and `gpt-3.5-turbo-0301` as ChatGPT. VERA in QA format actually means a T5 model finetuned on the same seen multiple-choice data as VERA.