# OpenReview forum: "Vera: A General-Purpose Plausibility Estimation Model for Commonsense Statements"
_EMNLP/2023/Conference — EMNLP 2023 Main_

### Official Review · Reviewer_HDaL · 2023-08-01

**Soundness:** 3

**Excitement:**

4: Strong: This paper deepens the understanding of some phenomenon or lowers the barriers to an existing research direction.

**Paper Topic And Main Contributions:**

This paper introduces VERA, a general-purpose plausibility estimation model for commonsense statements. The model is trained on a large-scale dataset consisting of correct and incorrect commonsense statements sourced from commonsense QA datasets and knowledge bases. VERA outperforms existing models in solving commonsense problems, filtering LM-generated commonsense knowledge, and detecting commonsense errors made by ChatGPT.

**Reasons To Accept:**

1. The VERA model for plausibility estimation is of significant importance to the NLP community. It serves as a versatile tool, capable of being employed as both a reward model and a classifier for intermediate steps in various NLP tasks.

2. The paper's structure is well-organized, offering a clear and easy-to-follow presentation of the research content.

3. VERA demonstrates superior performance compared to existing models in several tasks, even outperforming GPT-4 in certain scenarios.

**Reasons To Reject:**

1. The main papers' content feels incomplete, and I believe that the ablation analysis to investigate the effect of each training objective holds crucial importance. It would be more beneficial to move this section to the main paper for better understanding and comprehensive insights.

2. The training objectives appear somewhat complex, and the lack of details about hyper-parameters in the experiments is a concern. Combining three loss functions might prove challenging during training without explicit guidance on hyper-parameter settings.

**Reproducibility:**

3: Could reproduce the results with some difficulty. The settings of parameters are underspecified or subjectively determined; the training/evaluation data are not widely available.

**Reviewer Confidence:**

4: Quite sure. I tried to check the important points carefully. It's unlikely, though conceivable, that I missed something that should affect my ratings.

---

> ### Author Rebuttal · Authors · 2023-08-25
>
> Thanks for the insightful feedback! We really appreciate your recognition of our work’s importance to the community, and our model’s superior performance over state-of-the-arts.
>
> Due to space limits, we had to put many details in the Appendix, but we will move them back to the main text in our next paper update, using the extra page. Below is our response to your specific concerns:
> 1. **Ablation on training objectives:** We have already demonstrated the impact of training objectives in Appendix E.1, and experimental results can be found in Figure 5. We will move this into the main text in our next paper update.
> 2. **Hyper-parameters:** We report the hyperparameters in Appendix C and Table 8, including the weights of each component in the combined training objectives. Specifically, we use 1.0 for the binary classification loss, 1.0 for the multiple-choice loss, and 0.1 for the supervised contrastive loss. These values are obtained from some moderate hyperparameter tuning, and we did not do extensive search due to training cost. We will make sure the details about hyperparameter tuning are clear in the paper, so as to promote reproducibility.

---

### Official Review · Reviewer_HcPV · 2023-08-04

**Soundness:** 4

**Excitement:**

4: Strong: This paper deepens the understanding of some phenomenon or lowers the barriers to an existing research direction.

**Missing References:**

For future work Contextual commonsense inference:
https://ojs.aaai.org/index.php/AAAI/article/view/17521
An approach to score contextual commonsense assertions:
https://arxiv.org/pdf/2302.05406.pdf
A source on commonsense plausibility:
https://aclanthology.org/P16-1137.pdf


**Paper Topic And Main Contributions:**

The paper proposes utilizing a language model, trained on a dataset of commonsense assertions and assertions adapted from QA datasets, to give a plausibility score for commonsense statements.  The author's show that their proposed method has high accuracy and recall on a wide variety of benchmarks. The methods are simple to implement (some pre-processing of data, and simple training with 3 different objectives).

**Questions For The Authors:**

I think it was overall an excellent work. I think the scaling of the models should be put in the main body of the work later and can help inform people. With a t5-large you can decent performance in a considerably smaller model.

One question is will the code for pre-processing be released?

**Reasons To Accept:**

Some reasons to accept the work are the following:
* Excellent writing, very clear and to the point presentation of the problem of lack of verification for commonsense statements
* Wide variety of tests on a wider variety of datasets with different types of statements
* Excellent appendix with useful information particularly on how effective smaller models are for this same task
* The authors evaluate a variety of very large language models to test the verification of commonsense assertions which is good for the whole community


**Reasons To Reject:**

Some suggestions:
* The authors may want to include a simple baseline of a GPT-2 cross entropy of tokens. It is used in https://ojs.aaai.org/index.php/AAAI/article/view/17521 and I believe that it would be good to compare this type of score (thresholded possibly) with the scores of the model the authors propose.
* The work, although exhaustive in tests, is very simple in nature: it consists of putting a classifier head on a language model to score sentences. Although the combination of training objectives and the dataset preprocessing is novel, the classification head on a language model is not.
* The work does not touch on how context plays a part of commonsense facts. Contextual commonsense inference is an important problem (as commonsense information may shift through a narrative), and this work does not address this particular issue (it may be good to mention it for future work)
* The delta percentages in table 3 should be removed and usefulness should be replaced with the delta symbol.
* The argument that the multi-label classification improves performance is not very backed up by Figure 5. Additionally, the ablation studies should have some error margins and significance to see if they actually do anything. It seems elminating multi-class loss *improves* the accuracy in type 1 things so how are you arguing that it is useful?
* No amount of runs are given, if significance is claimed (line 641) you may want to back up this statement with more runs/actual significance tests.
* I could not find good backing up of why the 2 stage training is better than mixing the data or the inverse of the stages, please list this in the appendix (or make it clearer where this is)


**Reproducibility:**

5: Could easily reproduce the results.

**Reviewer Confidence:**

5: Positive that my evaluation is correct. I read the paper very carefully and I am very familiar with related work.

**Typos Grammar Style And Presentation Improvements:**

I could not find typos, but I was not very active in looking for them. Overall excellent work!

---

> ### Author Rebuttal · Authors · 2023-08-25
>
> Thanks for your valuable feedback and suggested additional references! We really appreciate your recognition of the diversity and coverage of our evaluation, and the values in our analyses and baselines.
>
> Below is our response to your concerns: (we will include the additional experimental results in the next paper update)
> 1. **Cross-entropy baseline:** Thanks for your suggestion! We experimented with this cross-entropy baseline, with GPT-3.5 (text-davinci-002), and below is the comparison with other methods, on the problem-solving evaluation:
> | Model | Seen |
> | --- | --- |
> | Cross-entropy (text-davinci-002) | 66.02 |
> | text-davinci-003 | 75.41 |
> | Vera-T5 | **85.51** |
>     * This cross-entropy method greatly underperforms the zero-shot version of GPT-3.5 as well as Vera.
> 2. **Simplicity of modeling:** Our model architecture is indeed simple and standard in classification setup. In early experiments we found it unhelpful to increase complexity (e.g., replacing the classifier head with MLP), so we favored simplicity for easier model distribution with HuggingFace. Despite the simplicity, our model is a strong, one-of-a-kind model for commonsense, and we believe it can be a useful resource for the NLP research community.
>     * We appreciate your recognition of novelty in our method.
> 3. **Contextualized commonsense:** We agree that commonsense statements are often contextualized, and that contextualized/defeasible commonsense inference is an important research problem. In this work, we focused on classifying *self-contained commonsense statements* (Sec 2, Line 120), because it is a problem with its own significance and challenge. We will discuss contextualized commonsense inference as future work in our next paper update, along with existing discussion of multi-sentence commonsense statements (Sec 7, Line 627).
> 4. **Table 3 symbols:** We measure “usefulness” as the increment in accuracy due to contextualizing on the commonsense knowledge, and the delta is the relative change in knowledge usefulness when we apply Vera filtering.
> 5. **Ablation on training objectives:** While eliminating multi-class loss increases performance by ~2% on boolean benchmarks in unseen (type 1) set, it decreases performance by ~1% on multiple-choice benchmarks in the unseen (type 1) set, and slightly harms performance on benchmarks in the seen set. As reported in Table 7, multiple-choice benchmarks have a majority in the unseen (type 1) set, so the overall performance is higher when including the multi-class loss.
> 6. **Significance:** We did one run per experiment. The gains are significant due to the large margin and large evaluation set. We additionally conducted a paired t-test on the problem-solving performance (Sec 5.1) between Vera-T5 and the best baseline, Flan-T5, and demonstrate that our gains are indeed statistically significant:
> | Eval Part | Gain (%) | n | t-statistic | p-value |
> | --- | --- | --- | --- | --- |
> | seen | +6.01 | 14779 | -19.10 | 1x10^-80 |
> | unseen (type 1) | +4.03 | 4420 | -6.79 | 6x10^-12 |
> | unseen (type 2) | +4.48 | 11002 | -12.64 | 1x10^-36 |
> 7. **Ablation on training stages:** Thanks for the great suggestion! We additionally experimented with your two suggested ablations: (1) mixing the data of Stages A and B and training in one single stage; (2) inverting the order of Stages A and B. These ablations underperform our best model. Below is their problem-solving accuracy, compared to the original Vera-T5:
> | Model | Seen | Unseen (type 1) |
> | --- | --- | --- |
> | Vera-T5 | **85.51** | **81.65** |
> | mixing | 84.17 | 81.33 |
> | invert | 84.78 | 80.18 |
> 8. **Open-sourcing:** All code, data and models will be made publicly available, and we believe they will be useful resources to the NLP research community.

---

### Official Review · Reviewer_FtwG · 2023-08-04

**Soundness:** 3

**Excitement:**

3: Ambivalent: It has merits (e.g., it reports state-of-the-art results, the idea is nice), but there are key weaknesses (e.g., it describes incremental work), and it can significantly benefit from another round of revision. However, I won't object to accepting it if my co-reviewers champion it.

**Missing References:**

Language Models (Mostly) Know What They Know Kadavath et al. 2022

**Paper Topic And Main Contributions:**

This paper proposes training a discrimination model called Vera for commonsense statements. The authors pre-train Vera on both commonsense question answering datasets and commonsense knowledge bases. The authors also propose to incorporate a combination of three objectives to train the model. After training, the authors show that when combined with some calibration techniques, Vera outperforms existing models that can be used for commonsense statement verification and can be used for filtering machine-generated commonsense knowledge.

**Questions For The Authors:**

N/A

**Reasons To Accept:**

1. The paper is overall well written and easy to follow.
2. The proposed method is technically sound and clearly explained/formulated.
3. Empirical results are promising: fine-tuning a T5 can outperform ChatGPT and GPT-4 is impressive. The authors also explore some applications and conduct some analysis for the proposed method.
4. The authors conduct experiments on many datasets/settings and many results are provided.

**Reasons To Reject:**

1. The idea of training a discriminative model to verify predictions from other models is not new. While the application on commonsense statement verification is new and meaningful, the technical novelty of the method is not very strong.
2. The experimental comparisons are not very solid because the calibration methods are only applied to Vera but not to other models, which can lead to a very biased comparison, especially no ablation study on the impact of calibration is done. Second, the LLM baselines including ChatGPT and GPT-4 are not very carefully engineered (no step by step reasoning of in-context learning is done). There also lacks some analysis on the impact of the proposed combination of training objectives.

**Reproducibility:**

3: Could reproduce the results with some difficulty. The settings of parameters are underspecified or subjectively determined; the training/evaluation data are not widely available.

**Reviewer Confidence:**

4: Quite sure. I tried to check the important points carefully. It's unlikely, though conceivable, that I missed something that should affect my ratings.

---

> ### Author Rebuttal · Authors · 2023-08-25
>
> Thanks for the insightful feedback! We appreciate your recognition of the soundness of our proposed method and the coverage of our experiments.
>
> Below are our responses, in which we try to clear up some misunderstandings:
> 1. **Technical novelty:** We show that adopting certain techniques (e.g., contrastive objectives, two-staged training, negative data augmentation) are critical in producing a strong commonsense discrimination model. This is supported by another reviewer who said “the combination of training objectives and the dataset preprocessing is novel.” Also please note that “the method is too simple” is one of the [deprecated heuristics](https://2023.aclweb.org/blog/review-acl23/#2-check-for-lazy-thinking) in the reviewing guidelines from ACL.
>     * Additionally, instead of focusing on intricate technical novelty, our main contribution consists of task formulation and open resources. We observe that the latest LMs fall short in generating correct commonsense statements and detecting incorrect ones. We formalize the task of commonsense plausibility estimation, gather diverse data sources and process them into declarative format, and present a strong commonsense model. All code, data and models will be made publicly available, and we believe they will be useful resources to the NLP research community.
> 2. **Calibration on baselines:** Our calibration is a post-hoc strategy and does not affect the task performance metrics we report in Sec 5, so *our experimental results and comparisons are valid and unbiased*. We briefly mentioned this in Sec 3.3 (Lines 347-350), and below we provide a detailed justification (which we will add to our next paper update):
>     * Applying our calibration method – temperature scaling – does not affect the relative order of plausibility scores assigned to a given set of statements.
>     * For tasks with multiple-choice questions (Sec 5.1), calibration does not affect the argmax prediction for the above reason.
>     * For commonsense knowledge filtering (Sec 5.2), calibration does not affect the TPR/FPR numbers at each corresponding decision point, again for the above reason, so the ROC curves are valid.
>     * For True/False judgment problems (Sec 5.1 and 5.3), calibration does not move the plausibility scores across the decision boundary. We use logit z=0.0 (or equivalently, plausibility score s=0.5) as the True/False boundary. A positive (or negative) logit remains positive (or negative) after applying the temperature.
> 3. **Ablation of calibration:** We reported the impact of calibration in Figure 2 and Appendix Figure 3. Under “Calibration Curve”, we plot the calibration curve for both the uncalibrated version (w/ faded color) and calibrated version (w/ saturated color) of the Vera model. As shown in Figure 3, applying calibration reduces the calibration error (ECE) from 7% down to 1% on seen datasets, and from 9% down to 2% on unseen datasets. We will add more discussion on the impact of calibration in our next paper update.
> 4. **ChatGPT/GPT-4 CoT baselines:** Thanks for the suggestion! We are adding the ChatGPT/GPT4 baseline with few-shot CoT, and report their problem-solving performance on seen benchmarks as below (and contrast with zero-shot ChatGPT/GPT-4 as well as Vera):
> | Model | Seen |
> | --- | --- |
> | ChatGPT (few-shot CoT) | 65.19 |
> | GPT-4 (few-shot CoT) | 74.96 |
> | ChatGPT (zero-shot) | 62.11 |
> | GPT-4 (zero-shot) | 72.35 |
> | Vera-T5 | **85.51** |
>
>     * We found that adding few-shot CoT increases the performance of ChatGPT/GPT-4 by about 3%, and they are still below Vera by a large margin. We will replace the zero-shot ChatGPT/GPT-4 baselines with this few-shot CoT version in our next paper update.
>     * The few-shot CoT template we use (for OpenAI ChatCompletion API) is:
> ```
> messages=[
>     # below is one demonstration. The statements and explanations are dataset-specific
>     {'role': 'user', 'content': 'Question: Based on commonsense knowledge, is the following statement correct? Please first generate an explanation and then answer yes or no.\nStatement: Beads is formed by clouds.'},
>     {'role': 'assistant', 'content': 'Beads of water are ... So the answer is yes.'},
>     # 4 more demonstrations, some with no as answer
>     ...
>     # below is the evaluated commonsense statement
>     {'role': 'user', 'content': f'Question: Based on commonsense knowledge, is the following statement correct? Please first generate an explanation and then answer yes or no.\nStatement: {statement}'},
>     # and we extract the second-last token (yes or no) from the next assistant response
> ],
> ```
> 5. **Ablation on training objectives:** We discuss the impact of training objectives in Appendix E.1, and experimental results can be found in Figure 5. We will add these results to the main text in our next paper update.

---

### Meta-Review · Area_Chair_yt5F · 2023-09-19

**Recommendation:** 4

**Metareview:**

The paper proposes a method, Vera, that estimates the plausibility of any given commonsense statement.  Vera is built by finetuning T5 on ~6M statements from two commonsense KBs and 19 commonsense QA datasets. The paper is well-written, and the author compares their method with several LLMs. VERA outputs a real-valued score in [0,1]. The idea of using Vera to detect errors made by ChatGPT is interesting.

Pros:

- The idea of automatically estimating the plausibility of a natural language statement with respect to commonsense knowledge is important and will be useful for the community.
- As agreed by all reviewers, the paper has promising results, especially Vera outperforming ChatGPT and GPT-4 in estimating plausibility.
- Vera is able to detect erroneous commonsense statements from generative LMs (ChatGPT).

Cons:
- The authors show that Vera as a filter or critic can be useful for reinforcement-based QA models (Table 3). However, the Area Chair thinks there can be a knowledge leakage since Vera is finetuned on these datasets. It will be more useful if the authors show it on the Unseen dataset.
- The authors do not explain why a real-valued score is used to estimate the plausibility. For example, it is difficult to understand the difference between 0.6 and 0.65 plausibility scores.  It would be interesting to see how the plausibility score relates to human understanding of commonsense plausibility.
- The error (quantitative) analysis is missing. An analysis with respect to different common sense knowledge dimensions (physical, social, etc.) would make the paper stronger.

There are a few useful results and analyses in the Appendix of the paper. Especially the ablation study should be included in the paper.

Finally, The AreaChair appreciate the authors' response to clarify the doubts of the reviewers.

---

### Decision · Program_Chairs · 2023-10-07

**Decision:**

Accept-Main

**Comment:**

The paper proposes a method, Vera, that estimates the plausibility of any given commonsense statement.  Vera is built by finetuning T5 on ~6M statements from two commonsense KBs and 19 commonsense QA datasets. The paper is well-written, and the author compares their method with several LLMs. VERA outputs a real-valued score in [0,1]. The idea of using Vera to detect errors made by ChatGPT is interesting.

Pros:

- The idea of automatically estimating the plausibility of a natural language statement with respect to commonsense knowledge is important and will be useful for the community.
- As agreed by all reviewers, the paper has promising results, especially Vera outperforming ChatGPT and GPT-4 in estimating plausibility.
- Vera is able to detect erroneous commonsense statements from generative LMs (ChatGPT).

Cons:
- The authors show that Vera as a filter or critic can be useful for reinforcement-based QA models (Table 3). However, the Area Chair thinks there can be a knowledge leakage since Vera is finetuned on these datasets. It will be more useful if the authors show it on the Unseen dataset.
- The authors do not explain why a real-valued score is used to estimate the plausibility. For example, it is difficult to understand the difference between 0.6 and 0.65 plausibility scores.  It would be interesting to see how the plausibility score relates to human understanding of commonsense plausibility.
- The error (quantitative) analysis is missing. An analysis with respect to different common sense knowledge dimensions (physical, social, etc.) would make the paper stronger.

There are a few useful results and analyses in the Appendix of the paper. Especially the ablation study should be included in the paper.

Finally, The AreaChair appreciate the authors' response to clarify the doubts of the reviewers.